# Dominant role of *CDKN2B/p15INK4B* of 9p21.3 tumor suppressor hub in inhibition of cell-cycle and glycolysis

Yong Xia[1,9], Yan Liu[1,9], Chao Yang[2,9], Diane M. Simeone[3,4], Tung-Tien Sun[5], David J. DeGraff[6], Moon-shong Tang[7], Yingkai Zhang [2] & Xue-Ru Wu [1,4,8 ✉]

Human chromosome 9p21.3 is susceptible to inactivation in cell immortalization and diseases, such as cancer, coronary artery disease and type-2 diabetes. Although this locus encodes three cyclin-dependent kinase (CDK) inhibitors (p15INK4B, p14ARF and p16INK4A), our understanding of their functions and modes of action is limited to the latter two. Here, we show that in vitro p15INK4B is markedly stronger than p16INK4A in inhibiting pRb1 phosphorylation, E2F activity and cell-cycle progression. In mice, urothelial cells expressing oncogenic *HRas* and lacking p15INK4B, but not those expressing *HRas* and lacking p16INK4A, develop early-onset bladder tumors. The potency of CDKN2B/p15INK4B in tumor suppression relies on its strong binding via key N-terminal residues to and inhibition of CDK4/CDK6. p15INK4B also binds and inhibits enolase-1, a glycolytic enzyme upregulated in most cancer types. Our results highlight the dual inhibition of p15INK4B on cell proliferation, and unveil mechanisms whereby p15INK4B aberrations may underpin cancer and non-cancer conditions.

[1] Department of Urology, New York University School of Medicine, New York, NY, USA. [2] Department of Chemistry, New York University, New York, NY, USA. [3] Department of Surgery, New York University School of Medicine, New York, NY, USA. [4] Department of Pathology, New York University School of Medicine, New York, NY, USA. [5] Department of Cell Biology, New York University School of Medicine, New York, NY, USA. [6] Department of Pathology and Laboratory Medicine and Department of Surgery, Division of Urology, The Pennsylvania State University College of Medicine, Hershey, PA, USA. [7] Department of Environmental Medicine, New York University School of Medicine, New York, NY, USA. [8] Veterans Affairs New York Harbor Healthcare System, New York, NY, USA. [9] These authors contributed equally: Yong Xia, Yan Liu, Chao Yang. ✉email: xue-ru.wu@med.nyu.edu

CDKN2B gene resides on human chromosome 9p21.3 and encodes p15[INK4B], a cell-cycle regulator that exhibits inhibitory activities toward cyclin-dependent kinases CDK4 and CDK6 (refs. [1,2]). Along with CDKN2A gene, which encodes p14[ARF] and p16[INK4A] via usage of separate first exons, and signals through p53 and pRb pathway, respectively[3], these three CDK inhibitors arranged in tandem on 9p21.3 constitute a prominent tumor suppressor hub that is frequently deleted or epigenetically silenced prior to immortalization of primary cultured cells[4–6] and in pre-tumor lesions[7,8] and a multitude of tumor types[3,4,9–11]. Unlike p14[ARF] and p16[INK4A] whose tumor-suppressive activities have been well documented, p15[INK4B] is often regarded as functionally equivalent to p16[INK4A] and a bystander during 9p21.3 deletion. These notions are, however, in stark contrast to the phenomena that (i) p15[INK4B] can be targeted for deletion independently of p14[ARF] and p16[INK4A][2,12]; (ii) the three CDK inhibitors can be down-regulated separately by promoter-specific methylation, cis-acting long non-coding anti-sense RNA (i.e., ANRIL and c-ANRIL)[13–15] or divergent upstream signals (i.e., p15[INK4B] by transforming growth factor beta (TGF-beta)[16,17]; (iii) p15[INK4B] becomes critically important in the absence of p16[INK4A] [18]; and (iv) in genetically engineered mice the loss of both p14[ARF] and p16[INK4A] fails to cooperate with oncogenes to elicit tumors in certain cell types, such as urothelium[19]. By analyzing p15[INK4B] in vitro, in vivo, and in human tumor cohorts, we present multiple lines of evidence here demonstrating p15[INK4B] as a markedly stronger tumor suppressor than p16[INK4A] via dual inhibition of cell cycle and aerobic glycolysis (Warburg effect)[20]. We provide detailed structural basis that sets p15[INK4B] apart from p16[INK4A] functionally and mechanisms whereby p15[INK4B] exerts its potent activities. Our findings highlight a dominant role of p15[INK4B] as a principal component of 9p21.3 locus in tumor suppression and the essential contributions of p15[INK4B] deficiency to tumor initiation. Additionally, our study reveals an entry point through which to study the elusive molecular bases of non-cancer diseases, such as coronary artery disease, type 2 diabetes, and glaucoma, all of which are prominently associated with 9p21.3 aberrations[9,21–23].

## Results and discussion

**Loss of p15[INK4B] triggers early-onset and highly penetrant tumors from mouse urothelial cells expressing oncogenic HRas.** Our initial motivation to investigate CDKN2B that encodes p15[INK4B] (p15 hereon for brevity) was because of the high frequency (up to 70%) of 9p21.3 loss in human urothelial tumors[11,24–26], but the complete failure of loss of CDKN2A (encoding p14[ARF] and p16[INK4A] or p14 and p16, respectively, hereon) to cooperate with oncogenic HRas (Q61L) to induce any urothelial tumor in mice[19]. We hypothesized that, instead of p14 and p16, p15 might play a more important role in urothelial tumor suppression in the context of oncogenic HRas, and that p15 loss might be more functionally reflective of 9p21.3 loss. To test this, we crossed our Upk2-HRas transgenic mice, in which the expression of oncogenic HRas was driven by urothelium-restricted murine Upk2 promoter (Fig. 1a)[19,27], with ubiquitous CDKN2B knockout mice, in which exon 2 of CDKN2B was replaced with a neo gene (Fig. 1a), which ablated p15 expression[28]. Intercrosses of the off-spring yielded several hypothesis-testable genotypes (Fig. 1b), which we followed for up to 10 months for urothelial tumorigenesis (Fig. 1c–f). With dipstick test of spot urine from live mice as a preliminary screening for hematuria (Fig. 1c, arrow), a salient clinical manifestation in human bladder cancer patients[29], we were able to identify potential bladder tumors that were primarily associated with HRas[WT/*]/p15[−/−] genotype (Fig. 1c, upper panel, arrow). Gross anatomy of the urinary bladders from a cohort of

mice sacrificed every 2 months beginning at 2 months of age ($n = 10$/time point/genotype) revealed markedly enlarged bladders (Fig. 1c, lower panel (B as in bladder)) with significantly increased bladder weights (Fig. 1d, upper panel) in the HRas[WT/*]/p15[−/−] group, but not in any other groups)), even though the body weights did not differ significantly (Fig. 1d, lower panel). Hydronephrosis was noted only in the HRas[WT/*]/p15[−/−] group, particularly in the 4-month age group (Fig. 1c, lower panel (K as in kidney)), due likely to bladder tumor overgrowth and blockage of the urinary tract outlet. Overall, in HRas[WT/*]/p15[−/−] mice, bladder tumors were confirmed microscopically in 20% of the mice by 2 months, 80% of the mice by 4 months, and 100% of the mice by 6 months (Fig. 1e, f). Bladder tumors were also detected in HRas[WT/*]/p15[WT/−] mice (e.g., heterozygous p15 deletion), albeit at a much lower frequency (30% by 10 months), later onset (beginning at 4 months) and smaller sizes (visible only microscopically). While a relatively small number of HRas[WT/*]/p15[+/−] (in which p15 was heterozygous) also developed bladder tumors, the tumor cells retained one p15 allele. We presently cannot rule out whether the remaining allele is inactivated by hypermethylation, ANRIL, or inactivation of the upstream regulators, such as TGF-β (see later). This is an important question that warrants further investigation. By histology, the p15[−/−] mice exhibited normal-appearing urothelium and the HRas[WT/*] mice urothelial hyperplasia (Fig. 1f). In contrast, the HRas[WT/*]/p15[−/−] mice harbored bladder tumors that were consistently papillary-looking, low pathological grade, and confined to the urothelial layer without breaching the basement membrane, thus being low-grade, non-invasive tumors (Fig. 1f). These data provide the first in vivo experimental evidence indicating that p15 loss and HRas activation are two highly cooperative events that are necessary and sufficient to initiate low-grade, non-invasive bladder tumor, the most prevalent, recurrent and expensive type of bladder tumor in human patients[30–32].

**p15 is a markedly more potent urothelial tumor suppressor than p16 in vivo.** Although we previously showed that the loss of CDKN2A failed to cooperate with oncogenic HRas to induce urothelial tumors[19], and we showed here that the loss of p15 and oncogenic HRas together are highly tumorigenic in urothelial cells (Fig. 1), these two studies were done at different time periods and settings. Additionally, since CDKN2A encodes both p16 and p14 (p19 in mice), it was necessary that we ascertain under the same experimental condition the relative potency of p15 versus p16 (instead of p16 and p19 together) in urothelial tumor suppression as well as the in vivo effects of their loss in urothelial tumor initiation. With this in mind, we carried out crosses between Upk2-HRas* transgenic mice and p16 knockout mice in which exon 1α of CDKN2A gene was replaced with a neo gene, which inactivates p16 but not alternatively transcribed p19 (Fig. 2a and ref. [33]). In parallel, we carried out de novo crosses between Upk2-HRas* transgenic mice and the aforementioned p15 knockout mice (Fig. 2a). From these crosses we obtained a number of different genotypes, but chose to focus on several genotypes that were most germane to define the in vivo differences between p16 deficiency and p15 deficiency in urothelial tumorigenesis, i.e., (i) wild-type (WT) mice that exhibit normal urothelium; (ii) single-allele HRas[WT/*] mice which we previously showed to consistently develop simple urothelial hyperplasia before 10 months of age[19]; (iii) HRas[WT/*]/p15[−/−] mice; and (iv) HRas[WT/*]/p16[−/−] mice (Fig. 2b). Western blotting verified the absent expression of p15 and p16 in the urothelial cells of the aforementioned group (iii) mice and group (iv) mice, respectively (Fig. 2c). Interestingly, there was an induction of p15 in HRas*-expressing urothelial hyperplasia, and more so in HRas*-

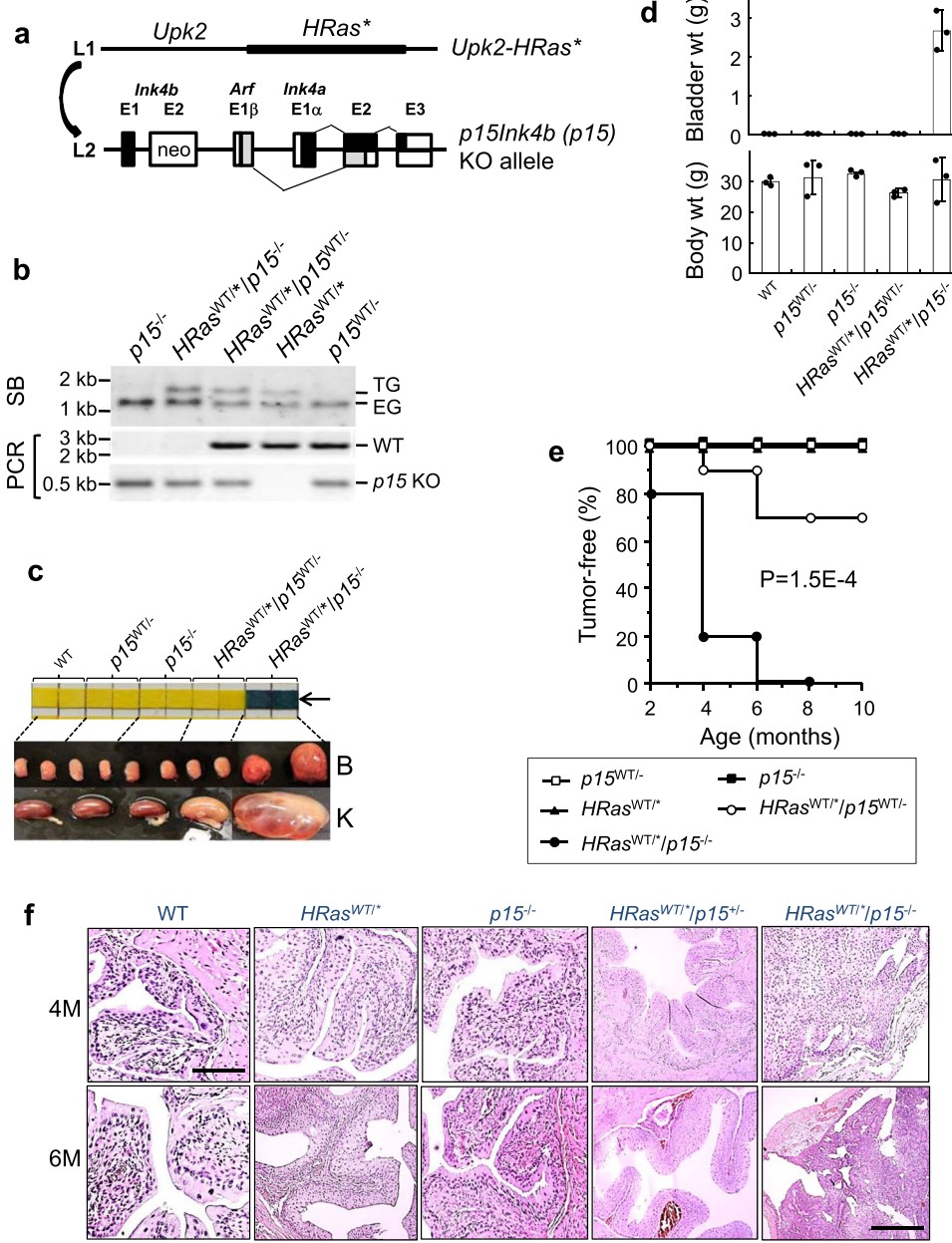

**Fig. 1 Loss of p15 in mouse urothelial cells expressing oncogenic *HRas* triggers early-onset and highly penetrant urothelial carcinomas that strongly resemble low-grade, papillary, non-muscle-invasive bladder cancer in humans. a** Mouse breeding scheme. Transgenic mice in which oncogenic *HRas* expression was under the control of murine *Upk2* promoter (abbreviated as *HRas*$^{WT/*}$ mice) was crossed and backcrossed to *p15Ink4b* knockout (*p15*$^{-/-}$) mice in which exon 2 of the gene was replaced by a neo gene, which ablates p15 expression[53]. **b** Representative genotypes from one litter of the aforementioned crosses. SB Southern blotting detecting the restriction enzyme-digested transgene fragment (TG) of *Upk2-HRas*, and endogenous *Upk2* fragment (EG); and PCR detecting the wild type (WT) and the *p15* knockout (KO) allele. Genotyping was performed on every mouse with highly reproducible results, and cumulatively *n* > 50/genotype. Source data are provided as a Source Data file. **c** Urine dipstick test of hematuria (upper panel; only the blood-testing row is shown) and gross anatomy of bladder (B) and kidney (K) (lower panel) of 4-month-old mice (2 mice/genotype shown). Color change to dark blue (arrow) indicates hematuria. Note the highly enlarged renal pelvis in the *HRas*$^{WT/*}$/*p15*$^{-/-}$ mice. The kidney of the *HRas*$^{WT/*}$/*p15*$^{WT/-}$ mouse was slightly bigger but lacked hydronephrosis upon sectioning. **d** Bladder and body weights (wt) of mice sacrificed at 6 months of age (3 mice/genotype). Data are presented as mean ± SD. **e** Tumor-free rate for genotypes indicated in the box (10 mice/time point/genotype). *P* value indicated (*P* = 1.5E−4) was for comparison between *HRas*$^{WT/*}$/*p15*$^{WT/-}$ mice and *HRas*$^{WT/*}$/*p15*$^{-/-}$ mice, using the Kaplan–Meier method followed by log-rank statistical test using software SPSS v17.0. **f** Representative H&E images of mouse bladders from the key genotypes at 4 and 6 months of age. *n* = 10/genotype/time point. All the panels are of the same magnification with the bar in the upper-left panel equaling to 100 μm, except the lower right panel (scale bar = 500 μm).

expressing and p16-lacking urothelial cells. While p16 was not significantly induced in *HRas**-expressing urothelial cells, deletion of p15 from these cells coincided with a strong induction of p16 (Fig. 2c). It appeared therefore that oncogenic *HRas* triggered a reciprocal compensatory induction of p15 and p16 in urothelial cells when the other protein is absent, albeit with divergent potencies in tumor suppression (see later). Finally, there was a strong increase of p19 expression that was confined to *HRas**-expressing and p15-lacking urothelial cells (Fig. 2c). However, such an increase clearly did not prevent urothelial tumor

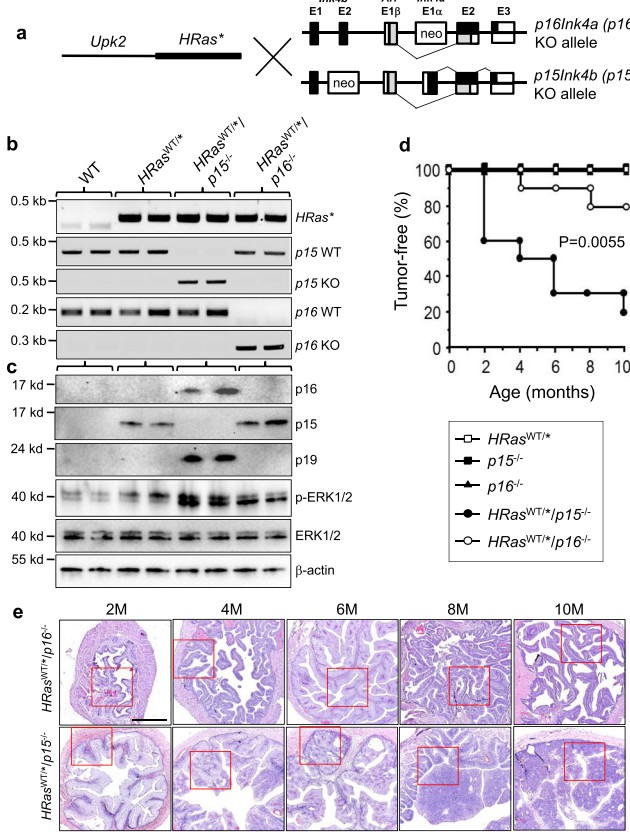

**Fig. 2 Divergent effects of p15 loss or p16 loss in *HRas*-mediated urothelial tumorigenesis. a** Mouse breeding scheme. Two sets of intercrosses were carried out (i) between *Upk2-HRas* transgenic mice (*HRas\**) and *p16Ink4a* (*p16*) knockout (*p16⁻/⁻*) in which exon 1α of the gene was replaced by the neo gene, thus leaving an independent transcript of the gene, *p19Arf*, intact; and (ii) between *HRas\** transgenic mice and *p15Ink4b* (*p15*) knockout (*p15⁻/⁻*) mice. **b** Representative genotyping results by genomic PCR of major experimental groups and controls (1 mouse/lane; 2 mice/genotype). The genotyping was performed on every mouse with highly reproducible results. **c** Western blotting of p16, p15, p19, ERK1/2 and phosphorylated ERK1/2, and β-actin (loading control) in urothelial cells of specific genotypes. Each lane represents one mouse and two independent mice were used per genotype. The experiment was repeated twice with similar results. Source data are provided as a Source Data file. **d** Divergent tumorigenesis in urothelial cells expressing *HRas\** and lacking p16 versus those expressing *HRas\** and lacking p15 ($n = 10$/time point/genotype). *P* value shown resulted from the comparison between *HRas^{WT/\*}/p16⁻/⁻* and *HRas^{WT/\*}/p15⁻/⁻* mice, using the Kaplan–Meier method followed by log-rank statistical test by software SPSS v17.0. Note the markedly earlier onset and higher frequency of tumorigenesis in *HRas^{WT/\*}/p15⁻/⁻* mice than in *HRas^{WT/\*}/p16⁻/⁻* mice. Also note that, while the significant induction of p16 in p15-lacking mice did not prevent bladder tumor formation, the significant induction of p15 in p16-lacking mice did exert a strong tumor-suppressive effect (Fig. 1d). **e** Bladder tumorigenic status in *HRas^{WT/\*}/p16⁻/⁻* versus *HRas^{WT/\*}/p15⁻/⁻* mice. Representative images of H&E-stained cross-sections of bladders from mice (age 2–10 months) expressing oncogenic *HRas* and lacking p16 or p15 as shown in the cohort in Fig. 1d. Note that the *HRas^{WT/\*}/p15⁻/⁻* mice developed nodular urothelial hyperplasia at 2 months of age and superficial papillary bladder tumors from 4 months onward, and that in contrast, the bladders of the *HRas^{WT/\*}/p16⁻/⁻* mice only contained mucosa folds from simple urothelial hyperplasia, and were largely devoid of tumors (see corresponding high-powered images for red-boxed areas in Supplementary Fig. 1). All panels are of the same magnification and the scale bar in the upper-left panel equals 500 μm. $n = 10$ mice per genotype per time point.

formation. Whether this prevented urothelial tumor progression, i.e., from the non-invasive to the invasive stage, requires further investigation.

Tumorigenesis analysis showed that *HRas^{WT/\*}* mice were devoid of any urothelial tumors, and that only 10 and 20% of *HRas^{WT/\*}/p16⁻/⁻* mice developed bladder tumors by 6 and 8 months of age, respectively (Fig. 2d), that could only be visualized microscopically. In contrast and general support of our initial cohort shown in Fig. 1, bladder tumors arose in 40% of *HRas^{WT/\*}/p15⁻/⁻* mice as early as 2 months, in 70% of the mice by 6 months and in 80% of the mice by 10 months (Fig. 2d, e and Supplementary Fig. 1). Without exception, these tumors were of low pathological grade, superficially located and papillary-looking (Fig. 2e; for higher magnification, see Supplementary Fig. 1). To further establish the different potencies of p15 versus p16 in tumor suppression in the context of *HRas\** mutation, we analyzed yet another cohort of mice (Supplementary Fig. 2). The data were highly reproducible in that, by 3 months of age, 40% of the *HRas^{WT/\*}/p15⁻/⁻* mice developed urothelial tumors, whereas none of *HRas^{WT/\*}/p16⁻/⁻* did the same. We do recognize the slight differences in tumor onset and frequencies in our *HRas^{WT/\*}/p15⁻/⁻* mice between Cohort 1 (Fig. 1e) and Cohorts 2 (Fig. 2d) and 3 (Supplementary Fig. 2). We believe that small differences in genetic background, i.e., Cohort 1 mice were closer to C57BL/6 and Cohort 2 and 3 mice were closer to into FVB/N, might contribute to those differences.

While the tumor cells in the *HRas^{WT/\*}/p15⁻/⁻* mice continued to express urothelial markers, such as Upk3a[34], they diminished in intensity and were accompanied by a significant increase of markers indicative of cell proliferation, such as Ki67, or urothelial tumor progenitor cells, such as Krt5 and Krt14 (Fig. 3a)[35,36]. The opposite was true in *HRas^{WT/\*}/p16⁻/⁻* mice (Fig. 3a). These data strongly suggest that, during *HRas\**-mediated urothelial tumorigenesis, p15 loss plays a much more critical role in bladder tumor initiation than p16 loss. Furthermore, while we observed a reciprocal induction of p15 and p16 in the absence of the other protein (Fig. 2c), deletion of both p15 and p16 in *HRas^{WT/\*}/Ink4ab⁻/⁻* mice did not significantly increase the tumor frequency or shorten the tumor latency in two independent cohorts (Supplementary Fig. 3) beyond what was seen in *HRas^{WT/\*}/p15⁻/⁻* mice (Figs. 1e, f and 2d, e and Supplementary Figs. 1 and 3). This further supports the idea that p15 loss is functionally more important than p16 loss in *HRas\**-mediated urothelial tumorigenesis, and that loss of both p16 and p15 does not appear synergistic or significantly additive in urothelial tumor initiation. Nevertheless, further investigation with large cohorts and more time points is necessary to firmly establish this conclusion.

**p15 is significantly more inhibitory of cell-cycle progression and proliferation than p16.** To understand the mechanisms underlying the different potencies of p15 and p16 in urothelial tumor suppression, we examined cell-cycle distribution by fluorescence-activated cell sorting (Fig. 3b) and pRB1 phosphorylation by western blotting (Fig. 3c) in the in vivo mouse cohort as shown in Fig. 2d, e and Supplementary Fig. 1. The fraction of urothelial cells in G0+G1 phases was much lower and the fraction in S and G2/M phases much higher in *HRas^{WT/\*}/p15⁻/⁻* mice than in *HRas^{WT/\*}/p16⁻/⁻* mice (Fig. 3b). This corresponded well with the elevated phosphorylation of pRB1 at both S780 and S807/811 sites in *HRas^{WT/\*}/p15⁻/⁻* mice (Fig. 3c), suggesting that the lack of p15 reduced the inhibition on CDK4 and/or CDK6 significantly more so than the lack of p16.

To independently verify the in vivo findings, we examined the potency of p15 and p16 on growth inhibition by restoring their

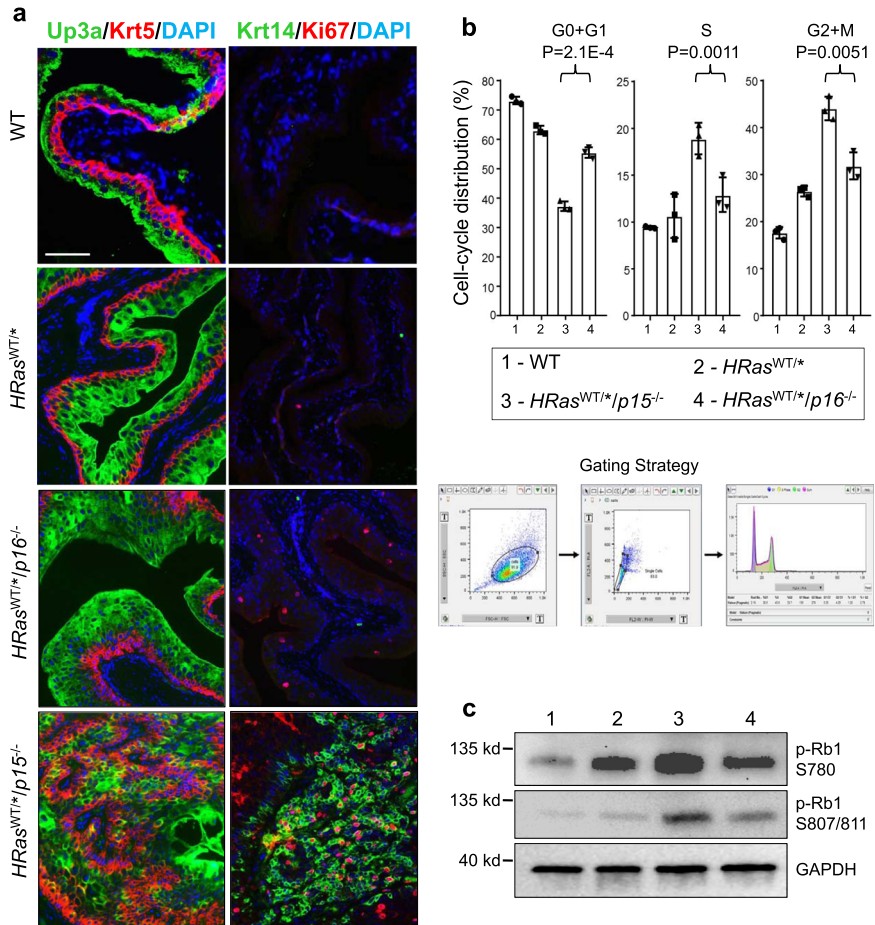

**Fig. 3 Comparison of proliferative status of urothelia in age-matched (all 6 months) wild type (WT), _HRas_^WT/*, _HRas_^WT/*/_p16_^−/−, and _HRas_^WT/*/_p15_^−/− mice. a** Immunofluorescent staining of urothelial differentiation (Upk3a), proliferation (Ki67), and bladder cancer progenitor cell (Krt5 and Krt14) markers. Note the significantly greater Krt5, Krt14, and Ki67 positive urothelial cells in _HRas_^WT/*/_p15_^−/− mice than in _HRas_^WT/*/_p16_^−/− mice. $n = 5$ per genotype indicated on the left. The panels are of the same magnification and the bar in the upper-left panel equals 50 µm. **b** Cell-cycle distribution (3 mice/genotype). Freshly stained urothelial cells by propidium iodine underwent (see Gating Strategy) (i) forward versus side scatter to remove small debris and large aggregates, (ii) width versus area to remove doublets, and (iii) cell-cycle distribution based on DNA content using FlowJo-embedded cell-cycle analysis. Data are presented as mean value ± SD. Two-sided _t_-tests were performed to compare the significance of the differences between the two groups (HRas^WT/*/p15^−/− versus HRas^WT/*/p16^−/−). The P values are shown in the figure. Note that urothelial cells in S and G2+M phases were 2–3-fold higher in _HRas_^WT/*/_p15_^−/− mice than in _HRas_^WT/*/_p16_^−/− mice. **c** Western blotting (1 mouse/lane) detection of phosphorylated pRb1. Source data are provided as a Source Data file. Note that the pRb1 phosphorylation (at both S780 and S807/811) was significantly higher in _HRas_^WT/*/_p15_^−/− mice than in _HRas_^WT/*/_p16_^−/− mice.

expression in two human bladder cancer cell lines (UMUC3 and RT112) that had homozygous deletion of 9p21.3 (ref. [37]) and therefore lacked endogenous p15 and p16. By first stably transfecting the cell lines with CMV-driven reverse tetracycline transactivator (CMV-rtTA) and then stably transfecting with tetracycline response element (TRE)-driven, Myc-tagged p15 cDNA or TRE-driven, Myc-tagged p16 cDNA, we were able to use tetracycline to inducibly express, detect and semi-quantify, via the Myc-tagged, restored p15 or p16. We found p15 to be significantly more inhibitory of cell proliferation than p16 in both cell lines (Fig. 4a). Additionally, compared with p16, p15 more significantly increased the fractions of cells in G0+G1 phases and decreased the fractions in S and G2+M phases (Supplementary Fig. 4a). Furthermore, p15 was significantly more inhibitory of pRB1 phosphorylation at S780 and S807/811 than p16, whereas neither p15 nor p16 affected the phosphorylation of other Rb family members pRB2 and p107 (Fig. 4b). Consistent with our results on reduced pRB1 phosphorylation by p15 expression which should lead to increased pRB1/E2F binding and decreased

E2F release, the overall E2F activity was much lower in p15-expressing cells than in p16-expressing cells (Supplementary Fig. 4b).

**p15 binds more efficiently to CDK4 and CDK6 than p16.** Because CDK inhibitors inhibit CDKs via direct binding[38,39], we tested whether p15 bound to CDK4 and CDK6 more efficiently than p16 using two independent approaches. In the first, we inducibly expressed Myc-tagged p15 or Myc-tagged p16 in UMUC3 cells, and carried out immunoprecipitation using an anti-Myc antibody, followed by western blotting detection and semi-quantification of CDK4 and CDK6 that naturally bound to p15 or p16. When normalized to immunoprecipitated Myc-tagged p15 and Myc-tagged p16 and then to input, p15 immunoprecipitated roughly sixfold and fivefold of CDK4 and CDK6, respectively, compared to p16 (Fig. 4c). In the second approach, we performed a pull-down experiment of freely available CDK4 and CDK6 in the total protein extracts by expressing

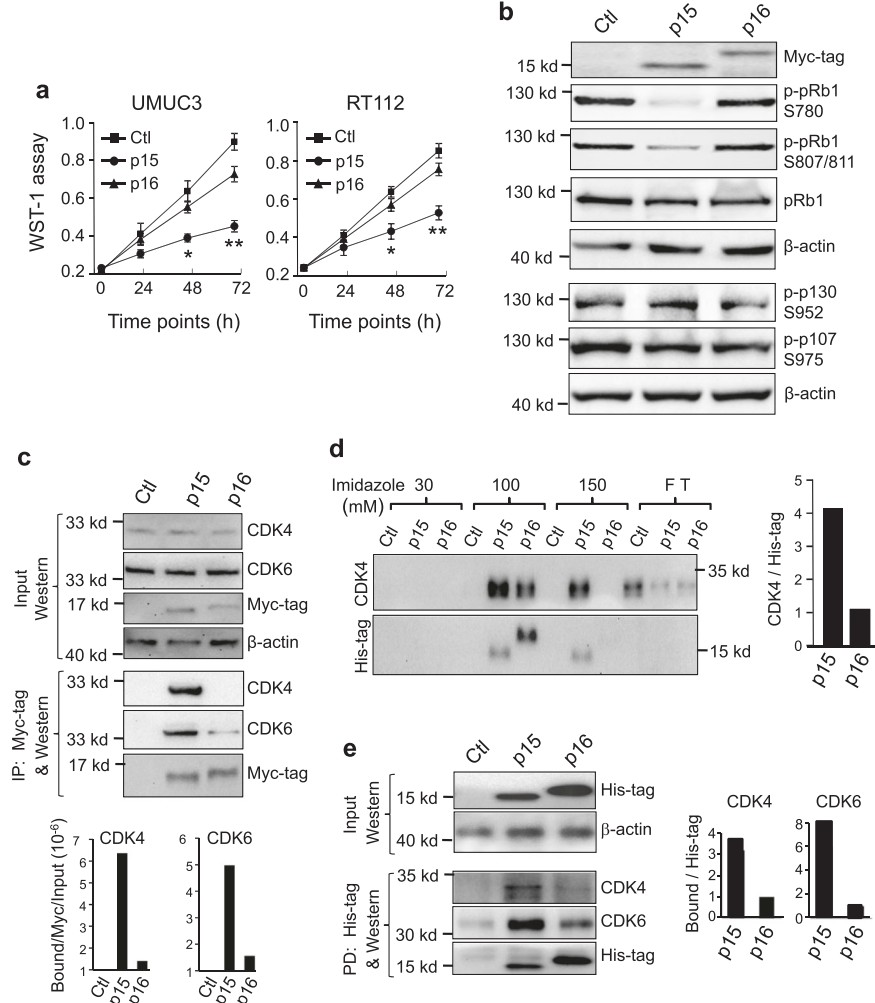

**Fig. 4 Greater inhibition of proliferation by p15 than by p16 via their differential binding to CDK4 and CDK6. a** Human bladder cancer cell lines, UMUC3 and RT112, both lacking endogenous p15 and p16, were stably transfected and induced to express either protein, using a two-vector, tetracycline-inducible system, followed by proliferation (WST-1 assay) at the indicated post-induction time points. Cell number was normalized by western blotting (as shown in **b**). In **a** $n = 4$ of biologically independent samples. Data are presented as mean value ± SD. Two-sided $t$-test was performed to compare the significance of the difference between the two groups (p15 versus p16). The $P$ values are: for UMUC3: (*), 1.3E-4; (**), 3.5E-5; and for RT112: (*) 0.0019 and (**) 8.8E-5. Note only a relatively small inhibition of proliferation and cell-cycle progression by p16 expression and a much greater inhibition by p15 expression. **b** Western blotting of key cell-cycle regulators of transfected UMUC3 cells analyzed 24 h after the induced expression of p15 or p16, showing the marked inhibition of pRb1 phosphorylation by p15 and much less so by p16, and that p15 did not affect the phosphorylation of other pRb family proteins p130 or p107. $n = 3$ per genotype indicated at the top. **c–e** Differential binding of p15 and p16 to CDK4 and CDK6. **c** Immunoprecipitation. UMUC3 cells were stably transfected and induced by tetracycline to express Myc-tagged p15 or Myc-tagged p16, followed by immunoprecipitation using anti-Myc-tag antibody and then western blot detection of co-immunoprecipitated proteins. The lower panels were representative semi-quantitation by densitometry of immunoprecipitated CDK4 and CDK6 by p15 or p16 in reference to the Myc-tag (middle panels) and input (top panels). The experiments were repeated three times with similar results. Note that, with equivalent amounts, p15 immunoprecipitated significantly more CDK4 and CDK6 than p16. **d** Pull-down assay. Histidine-tagged p15 or p16 produced by recombinant *E. coli* were immobilized on nickel columns and incubated with total protein extracts from untransfected, parental UMUC3 cells. Bound proteins were eluted by increasing concentrations of imidazole and detected by western blotting. The right panel was a representative semi-quantitation by densitometry of pulled-down CDK4 by p15 or p16 at 100 mM imidazole in reference to the His-tag. The experiments were repeated twice with similar results. Note that, proportionally, with 100 mM imidazole, p15 pulled down significantly greater amounts of CDK4 than p16. **e** An independent experiment similar to that described in **d**, showing greater amounts of CDK4 and CDK6 were pulled down by p15 than by p16. The right panels were representative semi-quantitation by densitometry of pulled-down CDK4 and CDK6 by p15 or p16 in reference to the His-tag. The experiments were repeated twice with similar results. Source data are provided as a Source Data file.

recombinant, histidine-tagged p15 or histidine-tagged p16 in *E. coli*, conjugating either protein to nickel columns, and incubating them with total proteins extracted from untransfected, parental UMUC3 cells. When referenced to histidine-tagged p15 and histidine-p16 eluted by imidazole at 100 mM, over fourfold more CDK4 was eluted with p15 than with p16 (Fig. 4d). This result was confirmed by another independently performed experiment, showing about four- and eightfold enrichment of CDK4 and

CDK6, respectively, with p15 as a bait, compared to p16 as a bait (Fig. 4e).

**The N terminus of p15 mediates the strong binding of p15 to CDK4 and CDK6.** p15 and p16 are highly homologous (69% identity in amino acid sequences) with the exception of their N-termini (N) and the longer C terminus (C) unique to p16. To discern the region responsible for the strong interaction between

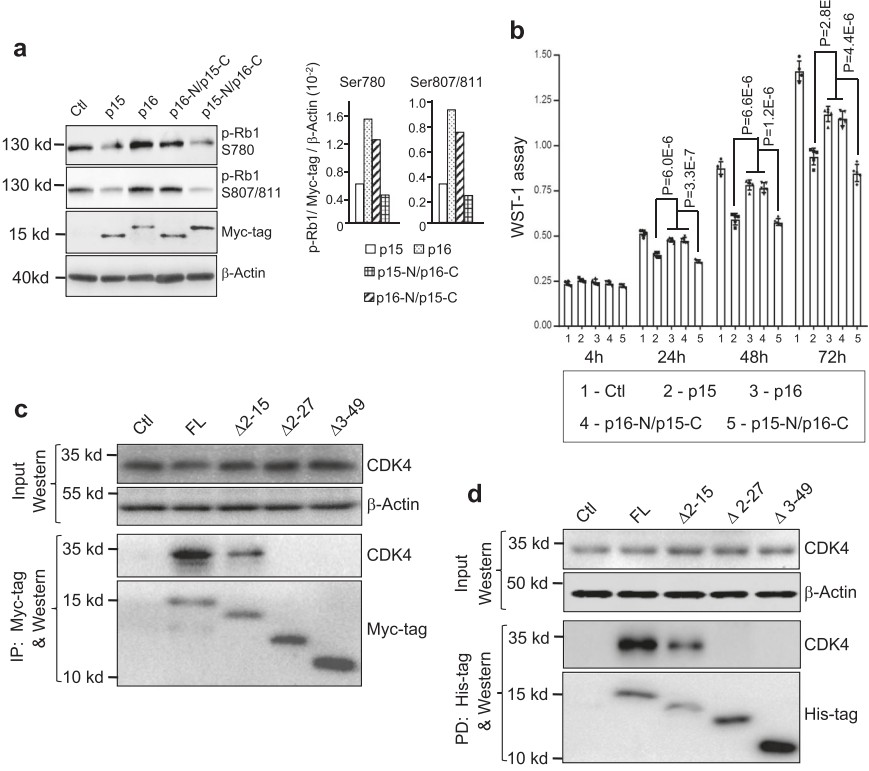

**Fig. 5 Critical role of the N terminus of p15 in binding to and inhibiting CDK4 and CDK6. a**, **b** Domain switching between p15 and p16 narrows strong growth inhibitory activity of p15 to its N terminus. Mammalian expression vectors bearing Myc-tagged cDNA hybrids (i.e., p16-N terminus (p16-N) plus p15-C terminus (p15-C)) or p15-N terminus (p15-N) plus p16-C terminus (p16-C)), along with control vectors (Ctl) bearing Myc-tagged wild-type p15 or wild-type p16 were stably transfected and inducibly expressed in UMUC3 cells. pRb1 phosphorylation status 24 h after induced expression of wild type and hybrid proteins (**a**) and cell proliferation with time course indicated (**b**) were assessed by western blotting with semi-quantitation and WST-1 assay, respectively (**a**, right panels), were representative semi-quantitation by densitometry of the phosphorylated pRb1 shown in the left panel in reference to Myc-tag and β-actin. The experiments were repeated twice with similar results. **b** p16 and p16-N/p15-C were statistically compared as one group because of their highly similar readings and compared with p15 and p15-N/p16-C as individual groups. Note that p15-N/p16-C was as effective as wild-type p15 in inhibiting pRb1 phosphorylation and cell proliferation and, in comparison, that p16-N/p15-C was as ineffective as wild-type p16 in inhibiting pRb1 phosphorylation and cell proliferation. $n = 5$ of biologically independent samples. Data are presented as mean ± SD. Two-sided $t$-test was performed to compare the significance of the difference between the two groups (p15 versus p16; and p16-N/p15-C versus p15-N/p16-C). The $P$ values are shown in the figure. **c**, **d** N-terminal deletion of p15 further narrows the critical region for CDK4 binding. Mammalian expression vectors bearing Myc-tagged deletion mutants of p15 cDNA as indicated above each panel, along with mock vector control (Ctl) and full-length (FL) p15 cDNA were stably transfected and inducibly expressed in UMUC3 cells. Immunoprecipitation (IP) followed by western blotting was then carried out (**c**). Alternatively, the histidine-tagged mutants were expressed in *E. coli*, immobilized on nickel column and incubated with total protein extracts from parental UMUC3 cell line (pull-down experiment, PD), followed by western blotting (**d**). Note that in both experiments the deletion mutant Δ2–15 reduced binding to CDK4 and the deletion mutants Δ2–27 and Δ3–49 completely abolished its binding ability to CDK4. Experiments in **c** and **d** were repeated three times with similar results.

p15 and CDK4/CDK6, we first performed domain switch between p15 and p16 by engineering p15/p16 hybrids, i.e., p15-N/p16-C and p16-N/p15-C. Transfection and inducible expression of Myc-tagged hybrids and their wild-type controls in p15/p16-lacking, parental UMUC3 cells showed that p15-N/p16-C was as effective as wild-type p15 in inhibiting pRB1 phosphorylation (Fig. 5a) and cell proliferation (Fig. 5b). In contrast, p16-N/p15-C was as ineffective as wild-type p16 in inhibiting pRB1 phosphorylation (Fig. 5a) and cell proliferation (Fig. 5b). Step-wise N-terminal deletion followed by transfection and immunoprecipitation (Fig. 5c) or by production of deletion mutants in *E. coli* and pull-down experiments (Fig. 5d) further showed that the deletion of the first 15 amino acids significantly reduced the binding of p15 to CDK4, and that the deletion of the first 27 amino acids completely abolished the binding of p15 to CDK4. These results established the critical importance of the N terminus of p15 in conferring its strong ability to bind CDK4 and CDK6 and inhibit pRb1 phosphorylation, cell-cycle progression, and proliferation.

**Key salt-bridges and hydrogen-bonds between p15 and CDK6 underlie strong binding.** Because the crystal structure of p15 is unavailable, we made use of the available structure of co-crystallized p16–CDK6[40] for computational modeling and molecular dynamics (MD) simulations for p15–CDK6 interaction (Fig. 6a, b, Supplementary Fig. 5 and Supplementary Table 3). The simulation results suggested several key interactive points between the N terminus of p15 and CDK6, particularly in the salt-bridges and H bonds (Fig. 6a). Specifically, the N-terminal residues Ser20 and Arg24 of p15 could form H bonds and salt-bridges with Glu18 and Glu21 of CDK6 (Fig. 6a). Additionally, Glu16 of p15 could form salt bridge with Arg45 of p15, which could stabilize Arg45 in a position to form additional salt bridge with Glu18 of CDK6 (Fig. 6a). These interactions could cooperate with each other to form a relatively stable network as suggested by MD simulations (Fig. 6a). In contrast, from both the p16–CDK6 crystal structure and p16–CDK6 MD simulations, p16 lacked such N-terminal interactions with CDK6 (Fig. 6a).

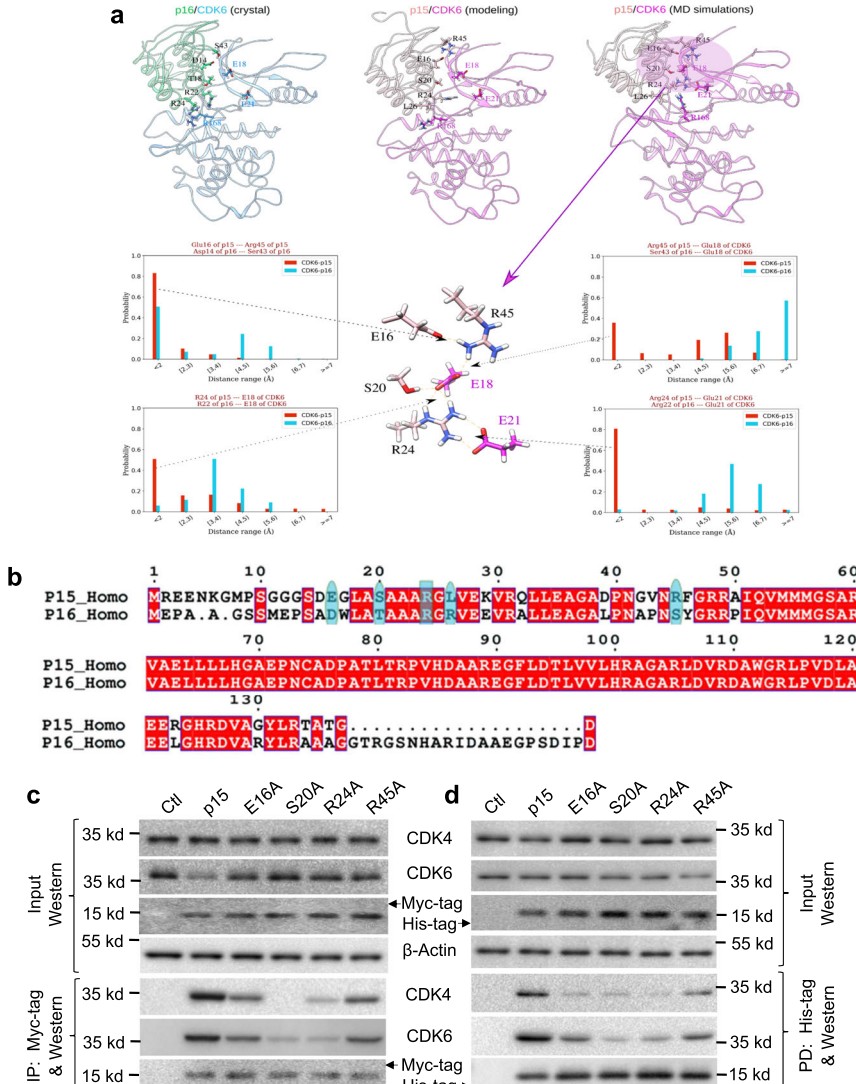

**Fig. 6 Structural modeling and experimental verification of amino acids in p15 that are critical for binding to CDK6. a** Homology modeling of p15 binding to CDK6 (top middle panel) based on the available crystal structure of p16/CDK6 complex (top left panel). Molecular dynamics simulations revealed that several N-terminal residues of p15 may form salt-bridges and H bonds with CDK6 (magenta circle of top right panel with enlarged simulation in the bottom middle panel), while the p16/CDK6 complex lacks such interactions (bottom left and right panels show the minimum distance distributions of identified key interactions for p15/CDK6, compared with p16/CDK6, in MD simulations). Specifically, Arg24 of p15 may be a hotspot residue in the p15/CDK6 interactions, and Glu16 of p15 may form a salt bridge with Arg45, which will be stabilized in the position and form salt bridge with CDK6. **b** Sequence alignment of p15 and p16 using Clustal W 1.83 (ref. [68]). **c**, **d** Site-directed mutagenesis of the putative CDK6-binding residues in the N terminus p15 to alanine followed by inducible expression of Myc-tagged proteins in UMUC and immunoprecipitation (**c**) or by pull-down using *E. coli*-produced, histidine-tagged mutant proteins in the presence of total proteins of UMUC cells (**d**). Note that the E16A, S20A, R24A, and R45A mutants had significantly reduced binding to both CDK4 and CDK6, although the mutational effects differed among the different mutants and binding detection methods (see "Results" for details). Experiments in **c** and **d** were repeated three times with similar results. Source data are provided as a Source Data file.

Using one-to-one residue mapping from sequence alignment of p15 and p16 (Fig. 6b), we found that Glu16, Ser20, Arg24, and Arg45 of p15 corresponded to Asp14, Thr18, Arg22, and Ser43 of p16, respectively. Structural and dynamics analysis of the p16–CDK6 complex indicates that, due to the repulsive force of Arg168 of CDK6 with Arg24 of p16 (corresponding to Leu26 of p15), the nearby Arg22 of p16 could not form salt-bridges with Glu18 and Glu21 of CDK6 (Fig. 6a). Our modeling therefore suggests a structural basis extending the experimental results indicating that p15 interacts more strongly with CDK6 than p16.

To experimentally validate our computational hypotheses, we carried out structurally guided site-directed mutagenesis by replacing Glu16, Ser20, Arg24, and Arg45 of p15 with alanine. Expression of these mutants fused with Myc-tag in UMUC3

followed by immunoprecipitation of CDK4 and CDK6 using anti-Myc antibody or by pull-down experiment using histidine-tagged mutants as baits and total protein extracts from parental UMUC3 cells showed that, compared to wild-type p15, the p15 mutants had markedly reduced interactions with both CDK4 and CDK6 (Fig. 6c, d). While the extent of weakened interactions was somewhat dependent on different mutants, methods of detection, and the interactions with CDK4 versus CDK6, the overall results were highly consistent with our simulation results regarding the key amino acids that were suggested to be critical for the binding between p15 and CDK6 (Fig. 6a, b).

To further test whether the relatively weak interaction between p16 and CDK6 was due in part to the repulsive force between Arg24 of p16 (corresponding to Leu26 of p15) and Arg168 of CDK6, we

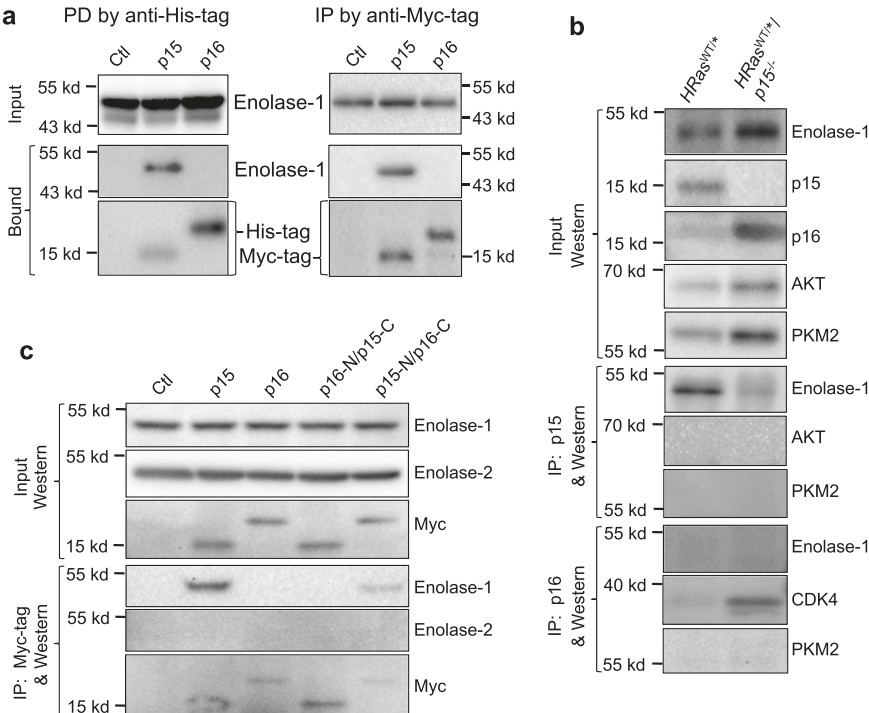

**Fig. 7 p15, but not p16, binds enolase-1 in vitro and in vivo. a** (left panel) Protein pull-down (PD) using *E. coli*-expressed, histidine-tagged and nickel column-immobilized p15 or p16 in the presence of total proteins from parental UMUC3 cells, and (**a**, right panel) immunoprecipitation (IP) of Myc-tagged p15 or p16, followed by western blotting. Note in both experiments the highly specific interaction of p15, but not p16, with enolase-1 with a molecular weight of 48-kDa. The side bars in the top two western blotting strips in the right panel denote molecular weight standards, 55-kDa (upper bar) and 43-kDa (lower bar). **b** p15 specifically interacts with enolase-1 in vivo. Total proteins were extracted from mouse urothelial cells expressing *HRas** or those expressing *HRas** and lacking p15, and were then subjected to immunoprecipitation followed by western blotting. One mouse per lane representative of a total of three independent mice per genotype. Note that enolase-1 was immunoprecipitated by anti-p15 antibody in p15-expressing, but not in p15-lacking urothelial cells (despite the fact that enolase-1 was significantly elevated in these cells); that p15 did not immune-precipitate AKT or PKM2; and that p16 did not immuno-precipitate enolase-1. **c** p15 binds, via its N terminus, to enolase-1. Myc-tagged, domain-switching mutants of p15 and p16 and wild-type controls were stably transfected and inducibly expressed in parental UMUC3 cells, and anti-Myc immunoprecipitation followed by western blotting was carried out. Note the specific immunoprecipitation of enolase-1, but not enolase-2, by p15 and p15-N/p16C, but not by p16 or p16-N/p15-C. Experiments in **a–c** were repeated three times with similar results. Source data are provided as a Source Data file.

replaced Arg24 of p16 with leucine, because MD simulations of p16–CDK6 complex showed that Arg22 of p16R24L could partially recover the salt-bridges with Glu18 and Glu21 of CDK6 (Supplementary Fig. 6a and 6b). Such a replacement indeed increased the binding between p16 and CDK6, but not with CDK4 (Supplementary Fig. 6c), suggesting that other residues in p16 that are divergent from p15 may also be less optimal for interacting with CDK4.

**p15 Interacts via its N terminus with enolase-1 in vitro and in vivo**. Having established that p15 is a more potent cell-cycle inhibitor than p16, we explored whether p15 played a non-canonical role in tumor suppression through interaction with other cellular protein(s). We took an unbiased pull-down approach using *E. coli*-produced, histidine-tagged p15 as a bait, with *E. coli*-produced histidine-tagged p16 or histidine-tag only as controls. After coupling the bait and controls separately onto nickel columns, we incubated them with total proteins extracted from parental UMUC3 cells. The bound proteins were eluted and resolved by two-dimensional electrophoresis, after which the proteins specifically bound to histidine-p15, but not to histidine alone or to histidine-p16, were selected for mass spectrometry (Supplementary Fig. 7). Of the 17 two-D gel spots analyzed, two belonged to enolase-1 (alpha enolase), one receiving a mascot score of 1823 with 117 peptide sequences identified by liquid chromatography tandem mass spectrometry (Supplementary Table 1).

To determine whether the interaction between p15 and enolase-1 was specific, we performed pull-down using *E. coli*-produced, histidine-tagged p15 and total proteins from parental UMUC3, as well as immunoprecipitation using inducible expression of Myc-tagged p15 in UMUC3 cells. Both experiments unequivocally validated the specific binding between p15 and enolase-1 (Fig. 7a). In contrast, under identical experimental conditions, p16 failed to bind enolase-1. p15, but not p16, also bound to enolase-1 in vivo in mouse urothelial cells expressing oncogenic *HRas* (Fig. 7b). Notably, p15-lacking/HRas-expressing urothelial cells had significantly elevated enolase-1 (Fig. 7b), which is consistent with upregulated glycolysis in tumor cells (see later). Additionally, whereas p15 bound to enolase-1, it did not bind to AKT or pyruvate kinase 2 (PKM2) (Fig. 7b), thus further demonstrating the binding specificity. Utilizing the aforementioned p15/p16 domain-switching mutants (as shown in Fig. 5), we further showed in enforced expression and immunoprecipitation experiments that it was the N terminus of p15 that mediated the interaction between p15 and enolase-1 (Fig. 7c). Finally, neither wild-type p15 nor p15-N/p16-C hybrid bound to enolase-2 (neuron-specific enolase) (Fig. 7c).

**Competitive binding among p15, enolase-1 and CDK4 and its effects on enolase-1 activity and aerobic glycolysis**. To assess whether the binding between p15 and enolase-1 reduces the

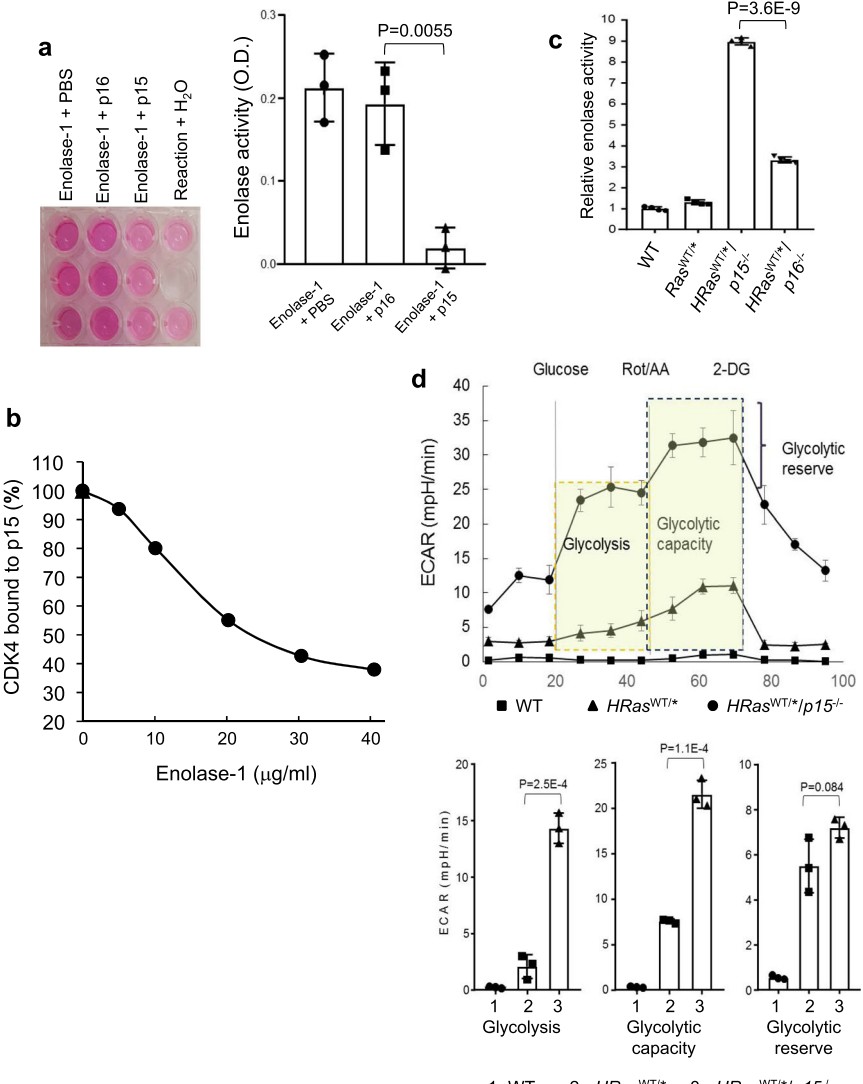

**Fig. 8 The competitive binding between p15, enolase-1, and CDK4/CDK6. a** p15, but not p16, inhibits enolase-1 activity in vitro. Purified enolase-1 was mixed with PBS, *E. coli*-produced p15 or *E. coli*-produced p16, and the enolase activity was measured using a commercial kit and expressed as relative optical density (OD). Note that p15, but not p16, inhibits enolase-1 activity. $n = 3$ of biologically independent samples. Data are presented as mean values ± SD. Two-sided *t*-test was performed to compare the significance of the difference between the two groups (p15 versus p16). The *P* values are shown in the figure. **b** Enolase-1 competitively inhibits p15 from binding to CDK4 in a dose-dependent manner. Purified enolase-1 was mixed with *E. coli*-produced p15, and the mixture was incubated with immobilized CDK4. Note that enolase-1 in a dose-dependent manner competitively inhibits p15 from binding to CDK4. $n = 3$ of biologically independent samples. Data are presented as mean values ± SD. **c** Enolase activity in relation to p15 and p16 status in vivo. Total proteins were extracted from urothelial cells of from WT, *HRas*^WT/* and *HRas*^WT/*/p15^−/− and *HRas*^WT/*/p16^−/− mice. Note the significantly elevated level of enolase activity in urothelial cells lacking p15. $n = 4$ of biologically independent samples. Data are presented as mean values ± SD. Two-sided *t*-test was performed to compare the significance of the difference between *HRas*^WT/*/p15^−/− and *HRas*^WT/*/p16^−/− mice. **d** Marked elevation of glycolysis in compound mice expressing *HRas** and lacking p15. Seahorse analysis was performed using urothelial cells from WT, *HRas*^WT/*, and *HRas*^WT/*/p15^−/− mice. $n = 3$ of biologically independent samples. Data are presented as mean values ± SD. Two-sided *t*-test was performed to compare the significance of the difference between the two groups (HRas^WT/*/p15^−/− versus HRas^WT/*). The *P* values are shown in the figure. Note that glycolysis, glycolytic capacity, and glycolytic reserve were all significantly elevated in *HRas*^WT/*/p15^−/− than in the controls (see lower panel for quantified comparisons).

catalytic activity of latter, we incubated the purified proteins in vitro, and found that p15, but not p16, significantly reduced the activity of enolase-1 (Fig. 8a). Because the N terminus of p15 mediates its binding to both CDK4 and enolase-1, we next examined whether the binding of enolase-1 to p15 could competitively inhibit CDK4 from binding to p15. In a plate assay, with increasing concentrations of enolase-1, we observed a dose-dependent decrease of the binding of CDK to p15 (Fig. 8b).

Consistent with the cell-free system, we found a strong correlation between p15 deficiency and elevated enolase activity in vivo. For instance, the enolase activity was about threefold higher in *HRas*^WT/*/p15^−/− mice than in *HRas*^WT/*/p16^−/− mice (Fig. 8c). Furthermore, by Seahorse analysis, the glycolysis, glycolytic capacity, and glycolytic reserve were all markedly elevated in *HRas*^WT/*/p15^−/− mice that lacked p15 expression (Fig. 8d). Together, our data reveal a role of p15 in regulating tumor cell metabolism via its interaction and inhibition of enolase-1 and glycolytic activities.

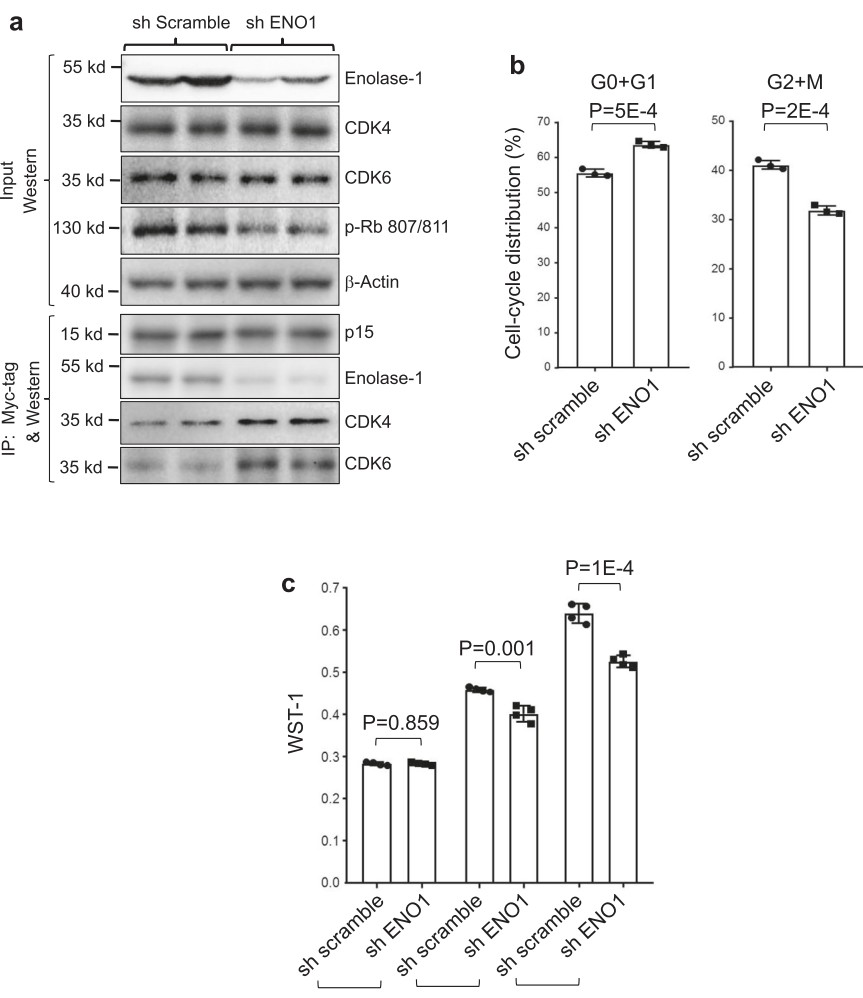

**Fig. 9 Down-regulation of enolase-1 reduces bladder cancer cell proliferation. a** knockdown of enolase-1 by shRNA in UMUC3 cells leads to increased binding of p15 to CDK4 and CDK6 as evidenced by immunoprecipitation followed by western blotting. The two lanes represented two stable clones from each of scrambled control and enolase-1 shRNA. Source data are provided as a Source Data file. **b**, **c** Reduced enolase-1 was associated with reduced cell-cycle progression and increased cell proliferation. $n = 3$ and 4 for **b** and **c**, respectively, of biologically independent samples. Data are presented as mean values ± SD. Two-sided $t$-test was performed to compare the significance of the difference between the two groups (p15 versus p16). Experiments were repeated three times with similar results.

**Down-regulation of enolase-1 suppresses bladder cancer cell proliferation**. To begin to address the biological effects of eno-lase-1, we down-regulated its expression using shRNA knock-down in UMUC cells expressing p15 (Fig. 9a). Compared to the shamble controls, the reduction of enolase-1 was associated with an increased binding between p15 and CDK4 and CDK6 (Fig. 9a) and reduced cell-cycle progression (Fig. 9b) and cell proliferation (Fig. 9c), thus lending further support to competitive nature observed in vitro of the binding among p15, CDK4/6, and enolase-1. These data further suggest that p15 plays a dual role in suppressing cell-cycle progression and glycolysis.

The findings we presented in this paper are highly significant on several fronts. First, the multiple lines of evidence we obtained in vitro and in vivo establish *CDKN2B* as a central player, rather than a bystander, of 9p21.3 locus in urothelial tumor suppression and the dominant role of *CDKN2B* deficiency in driving the formation of low-grade non-invasive bladder cancer. In doing so, we resolve a major discrepancy between the high prevalence of 9p21.3 deletion (~70%) along with RTK-*Ras* activation (~95%) in human bladder tumors[7,31,41] and the lack of tumorigenicity in mice by activated RTK-*Ras* pathway components alone or in combina-tion with *CDKN2A* deletion[19,42,43]. Our data strongly suggest that

*CDKN2B* deficiency is more reflective of 9p21.3 loss and is a functionally relevant collaborative partner with RTK-Ras activation to initiate bladder tumors. Consequently, the status of *CDKN2B*, instead of *CDKN2A*, coupled with RTK-*Ras* activation should be used as a combinatorial biomarker set to diagnose and predict the recurrence of low-grade non-invasive bladder cancer. Because concurrent 9p21.3 inactivation and RTK-*Ras* activation are highly prevalent in non-bladder cancer types[8,9,13], our findings call attention to the need to reexamine the effects of *CDKN2B* deficiency on tumorigenesis of other organ systems and cell types. Even greater attention should be paid to the tumorigenic role of *CDKN2B* deficiency in tumors where *CDKN2B* is preferentially deleted, such as familial melanoma, glioma, and certain hemato-logical malignancies[2,12,44,45]. Second, our data reveal *CDKN2B* as an important player that interconnects cell cycle and glycolysis by specifically binding and inhibiting both CDK4/CDK6 and enolase-1. Thus, the deficiency of *CDKN2B* not only promotes tumor cell proliferation by releasing the constraints of *CDKN2B* on G1 to S progression, it also creates a metabolically conducive environment during tumor-induced hypoxia under which tumor cells require increased glycolysis to produce metabolic intermediates for biosynthesis of cell building blocks, such as amino acids, lipids,

and nucleic acids[46]. By compiling gene expression data (Metabolic gEne RApid Visualizer (http://merav.wi.mit.edu/) comprising eight normal bladder cohorts and nine bladder tumor cohorts from human patients (https://www.cbioportal.org/), we found enolase-1 to be significantly overexpressed in all the tumor cohorts irrespective of tumor grades, compared to the normal controls (Supplementary Fig. 8). These data are consistent with two earlier microarray-based gene expression cohorts that had information on tumor stages, showing that enolase-1 is considerably higher in Ta-T1 non-muscle-invasive bladder tumors as well as in T2-T4 muscle-invasive tumors than in normal bladder specimens (Supplementary Fig. 9)[47,48]. It is worth noting that, within the muscle-invasive tumors, enolase-1 is generally the only overexpressed enolase isoform, instead of enolase-2 and enolase-3 (Supplementary Fig. 9). This is in agreement with the fact that normally enolase-2 and enolase-3 are primarily expressed in neurons and muscle cells, respectively, and are probably not used as the principal enolases for metabolism by normal urothelial cells. The specific upregulation and the unopposed activity of enolase-1 in the absence of CDKN2B in bladder tumor cells should open an avenue for therapeutic inhibition. Given the possibility that, even when CDKN2B is wild-type, enolase-1 could competitively inhibit it from binding to CDK4 and CDK6 (Figs. 8 and 9), blocking enolase-1 could potentially release CDKN2B and restore its activity as a cell-cycle inhibitor. Reprogramming the way by which glucose is utilized in bladder tumor cells by shifting glycolysis to oxidative phosphorylation may help stave off hypoxia-induced tumor cell proliferation and reduce the growth of low-grade superficial papillary bladder tumor growth and/or prevent it from recurring. The feasibility to deliver chemical agents directly into the bladder via a trans-urethral catheter should help attain the highest possible local concentration with lowest possible systemic toxicity[49]. Finally, our data should have implications beyond bladder cancer and cancers in general, because 9p21.3 is the most prominent susceptibility locus that is strongly associated with patients with coronary artery disease[21,50,51] and with type 2 diabetes[22,23]. While these sequence variants, i.e., rs10757278-G for coronary artery disease and rs10811661-T for type 2 diabetes, are located outside the coding region of CDNK2B, they have been shown to affect the activity of ANRIL, which in turn could downregulate CDNK2B expression[52]. Data from our present study regarding CDKN2B as a potent dual inhibitor of cell cycle and glycolysis should open new avenues to establish the disease linkage on a functional level and elucidate the cellular mechanisms whereby 9p21.3 aberrations lead to coronary artery disease, type 2 diabetes, and glaucoma.

## Methods

**Genetically engineered mice**. Upk2-HRas is a transgenic mouse line originally developed in the corresponding author's laboratory in which a 3.6-kB murine uroplakin II promoter drives urothelium-restricted expression of a constitutively active rabbit HRas oncogene (Q61L)[27]. Heterozygous Upk2-HRas which has been successfully maintained to date harbors a single-copy Upk2-HRas transgene and reproducibly develops simple urothelial hyperplasia of the bladder[19]. This line serves as breeders and a reference line representing tumor precursor lesion of the urothelium. CDKN2B (p15) knockout mouse line (a kind gift of Dr. Angel Pellicer of New York University School of Medicine) is a ubiquitous p15 knockout line in which the exon 2 of p15 gene was replaced with a neo, which abrogated p15 expression[53]. p16 knockout mouse line, obtained from US National Cancer Institute's Mouse Repository, is a ubiquitous p16 knockout line in which the exon 1α of CDKN2A gene was replaced with a neo gene, which inactivated p16 but not alternatively transcribed p19 (ref.[33]). Ink4ab (e.g., p15/p16) double knockout mouse line (a kind gift of Dr. Anton Berns of The Netherlands Cancer Institute) is a ubiquitous knockout line that was originally obtained in Dr. Berns's laboratory by targeting p15 in the p16 knockout ES cells where exon 2 of CDKN2A bore a stop codon in the p16 reading frame but not in the p19 reading frame[18]. Together, this abrogated the expression of p15 and p16 without affecting the p19 expression. Intercrosses were carried out between the heterozygous Upk2-HRas line and each of aforementioned knockout (KO) lines, and the offspring was further intercrossed such that the Upk2-HRas transgene was heterozygous and the KO alleles were hetero- or homozygous for tumorigenic analyses. The Upk2-HRas transgene allele

was identified by genomic Southern blotting of NcoI-digested tail DNA or by genomic PCR using specific primers (Supplementary Table 2). Knockout alleles and wild-type control alleles for p15, p16, and Ink4ab (p15/p16) were also verified by genomic PCR using specific primers (Supplementary Table 2).

All animal-related procedures were carried out in accordance with local and federal regulations and after the approval of a protocol by Institutional Animal Care and Use Committee (IACUC) of New York University School of Medicine.

### Urothelial tumorigenesis analyses

*Urine dipstick test*. Preliminary screening of hematuria, a common manifestation of bladder tumors in human patients, was performed on spot urine samples collected from conscious mice through gentle massaging of the lower abdomen. Urine samples were immediately applied atop of dipstick strips (BD Biosciences). The color change from yellow to blue on the sixth square from the top was considered positive for hematuria and the result was recorded on a digital camera.

*Gross anatomy and histopathology*. Mice of unique genotypes comprising different cohorts were sacrificed at pre-determined time points, and their urinary bladders were freshly dissected out, urine drained, weighed, turned inside out for the presence of visible tumors. The bladders were then bisected along the sagittal plane, with one half fixed in 10% PBS-buffered formalin, routinely dehydrated in serial ethanol and xylene, embedded in paraffin, sectioned, and stained with hematoxylin and eosin, and the other half scraped off its mucosa/tumors for protein preparation and biochemical work.

### Bladder cancer cell lines, stable transfection, and inducible gene expression.

Two human bladder cancer cell lines, RT112 and UMUC3, that had 9p21.3 deletion[37] were purchased from the American Type Culture Collection (Manassas, VA, USA) and used for all transfection experiments. Both the cell lines were cultured in Dulbecco's modified Eagle's medium (DMEM) (Gibco, Grand Island, NY, USA) supplemented with 10% Tet-system approved fetal bovine serum (Clontech, Mountain View, CA, USA) at 37 °C with 5% $CO_2$.

A tetracycline-based inducible system was chosen for conditional gene expression. Briefly, plasmid pTet-on (Clontech, Mountain View, CA, USA) was transfected into RT112 and UMUC3 cells using Lipofectamine 2000 (Invitrogen, Carlsbad, CA, USA), followed by selection of stable clones in 300 μg/ml G418 (Gibco, Grand Island, NY, USA). pTRE2-pur containing p15, p16, p15-N/p16-C or p16-N/p15-C was then transfected into pTet-on-stably transfected RT112 or UMUC3 using Lipofectamine 2000 and stable clones were selected in the presence of puromycin (Gibco) (0.8 μg/ml for RT112 and 1.2 μg/ml for UMUC3).

### Cell cycle and proliferation analyses

*Proliferation assay*. Cultured bladder cancer cells ($5 \times 10^3$) were seeded onto 96-well plates in DMEM medium with FBS containing 1 μg/ml doxycycline, and cell proliferation was determined at 24, 48, and 72 h by WST-1 (Clontech, Mountain View, CA, USA) with a 96-well plate reader at 450 nm absorbance (Dynex, Chantilly, VA, USA).

*Fluorescence-activated cell sorting*. Cultured cells were detached by trypsin digestion, collected, and washed by centrifugation at $800 \times g$ for 5 min. The cells were then fixed in pre-cooled, 70% ethanol, washed in PBS, and stained with propidium iodide/RNase staining solution (Cell Signaling Technology, Danvers, MA, USA) and sorted with a FACSCalibur analyzer, and the data were analyzed by FlowJo software (FlowJo 10.4.2).

*E2F activity assay*. E2F activity was assessed by transient transfection of UMUC3 cells with an E2F-luciferase reporter construct (Cignal E2F Reporter Assay Kit, Qiagen). Briefly, UMUC3 cells that reached 60% confluence were transfected the E2F reporter plasmid in the presence of Lipofectamine 2000 (Invitrogen, carlsbad, CA, USA). After 12 h of incubation, the transfected cells were lysed, supplemented with dual-luciferase reporter assay kit (Promega), and the luciferase activity was measured using a luminometer (Synergy H4 Hybrid Reader, Biotek,Winooski, VT, USA).

### Computational and MD simulation of p15–CDK6 interactions

*Computational modeling*. Because the crystal structure of p15 is unavailable, the structure of p16–CDK6 complex (PDB id: 1bi7)[40] was used as a template to model the molecular interactions between p15 and CDK6. The existing p16–CDK6 model missed two structural parts: one in CDK6 (residues 49–71) and the other in the N terminus of p16 (residues 1–9) (Supplementary Fig. 5a). The Modeler Program[54] was used to model these two missing parts. For the non-terminal missing part of CDK6, a highly similar CDK6 structure from the p18–CDK6–K-cyclin complex (PDB: 1g3n)[55] was used as the template to build the missing part (Supplementary Fig. 5b). For the missing part of the N terminus of p16, no template was used in the modeling procedure. For the p15–CDK6 complex, protein modeling of p15 (sequence 1–138) was performed on the web server of I-TASSER[56,57] following the standard instructions. The identified structural templates from the PDB database were p16 from the p16–CDK6 complex (PDB id: 1bi7)[40] and p15 from p15 NMR

ensemble (PDB id: 1d9s)[58]. Among the five p15 models predicted by I-TASSER based on p16, the one with the highest *C*-score was selected and superimposed on the p16–CDK6 complex to build p15–CDK6 complex (Supplementary Fig. 5d).

*MD simulation*. Two comparable systems: p16–CDK6 and p15–CDK6 complexes as well as an additional p16(R24L)–CDK6 complex were investigated in this study. For each system, five independent MD replicas were carried out using same starting structural configuration, but with different initial velocities. Total simulation time of 2.25 μs was performed (Supplementary Table 3). All MD simulations were carried out with Amber 16 package using Amber FF 14SB force field[59,60]. AmberTools[61] were used to create topology and coordinate files for the structures. Each system was neutralized by adding counterions of $Na^+$ or $Cl^-$, and was solvated in rectangular water box full of TIP3P water, so that the boundary of the box is at least 12 Å away from any solute atom. The cutoff distance was set at 10 Å for short-range Coulomb and van der Waals interactions, and the particle mesh Eward (PME) method[62] was utilized to treat long-range electrostatic interactions. The SHAKE algorithm[63] was employed on all atoms covalently bonded to hydrogen atom, allowing for an integration time step of 2 fs. The simulation detail is as follows. Each system was minimized by three steps: (i) the system was minimized by 3000 steps of the steepest descent minimization followed by up to 2000 steps of conjugate minimization, and harmonic restraints with a force constant of 5 kcal/mol Å were applied to all solute atoms; (ii) the above minimization procedure was repeated and harmonic restraint with a force constant of 1 kcal/mol Å imposed on solute atoms; and (iii) no harmonic restraint was imposed on solute atoms during the minimization. After minimization, each run was subjected to equilibration in following steps: (i) 50 ps constant volume ensemble (NVT) MD simulations with 2 kcal/mol Å restraints on the solute atoms, and the system was gradually heated from 10 to 300 K. (ii) 50 ps NVT MD at 300 K and with 2 kcal/mol Å restraints on the solute atoms. (iii) 100 ps isothermal isobaric ensemble (NPT) MD at 300 K and with 0.5 kcal/mol Å restraints on the solute to adjust the solvent density. Finally, production MD simulations were carried out for 150 ns at a constant temperature of 300 K and a constant pressure of 1 atm. The temperature and pressure were maintained using Langevin thermostat and Berendsen barostat methods, respectively. The atomic coordinates of the complexes were saved every 100 ps to obtain the trajectories for post structural analysis.

*Post structural analysis of MD trajectory*. The root-mean-square deviation (RMSD) profiles of protein C alpha atoms (Supplementary Fig. 10a) and B-factor analysis (Supplementary Fig. 10b) for p15–CDK6, p16–CDK6, and p16[R24L]–CDK6 complexes in different MD replicas were performed. For each MD replica, the residue B-factor analysis was conducted on the last 50-ns trajectories using *atomicfluct* command in cpptraj module of AmberTools[61]. The residue *B*-factor is the mass-weighted average of positional fluctuations squared of all atoms in each residue multiplied by a constant $8\pi^2/3$. For each system, a combined 500-ns trajectories (5 MD replicas of last 100-ns trajectories) were aligned and clustered. The clustering was performed on the binding interface residues by using the DBSCAN method[64] with min-points of 50 and epsilon of 1.2 Å, which is implemented in cpptraj module of AmberTools[61]. For the p15–CDK6 system, nine clusters were generated and the most populated one (832 out of 5000 snapshots) was selected for p15–CDK6-binding interface analysis (Fig. 6a). Besides the clustering analysis, the minimum distance distributions of identified key interactions were carried out over combined 500-ns trajectories for p15–CDK6, compared with corresponding residue pairs in p16–CDK6 (Fig. 6a), as well as those in p16[R24L]–CDK6 (Supplementary Fig. 6b).

## Construction of expression vectors containing wild-type cDNA, hybrids, deletion, and point mutations

*Myc-tagged p15 and p16*. To facilitate detection and minimize artifacts with the use of different antibodies for immunoprecipitation and western blot-based semi-quantification, a Myc-tag was inserted in-frame at the C terminus of wild-type p15 and p16 cDNAs (Addgene, Watertown, MA) by PCR using Q5 High-Fidelity DNA Polymerase (New England BioLabs), with p15- and p16-specific primers (Supplementary Table 2). The PCR products were then purified, sequence-verified, and cloned into pTRE2pur vector (Clontech) for transfection experiments.

*p15/p16 domain-switch constructs*. Two different hybrids, one comprising the N terminus of p15 and the C terminus of p16 (p15-N/p16-C) (1-methionine to 51-isoleucine of p15/52-glutamine to 156-aspartate of p16) and another comprising the N terminus of p16 and the C terminus of p15 (p16-N/p15-C) (1-methionine to 49-isoleucine of p16/51-isoleucine to 138-aspartate of p15) were constructed using the following procedures. First, PCR was carried out using pTRE2pur-p15-Myc as a template using primer 1 (see Supplementary Table 2 as with all other primers) and primer 2. Second, PCR was carried out using pTRE2pur-p16-Myc as a template, using primer 3 and primer 4. Third, the products from the above two steps were mixed and run for 10 PCR cycles, and then primer 1 and primer 4 were added and 30 additional PCR cycles were run. These steps yielded p15-N/p16-C. To generate p16-N/p15-C, pTRE2pur-p16-Myc was used as a template for PCR using primer 5 and primer 6. This was followed by PCR using pTRE2pur-p15-Myc as a template using primer 7 and primer 4. The PCR products from the above two steps were mixed and run for 10 PCR cycles. Primer 5 and primer 4 were then added and 30 additional PCR cycles were carried out. These steps yielded p16-N/p15-C. The p15-

N/p16-C and p16-N/p15-C cDNA hybrids were cloned separately into pTRE2pur vector for transfection experiments.

*Construction of histidine-tagged p15 and p16 and recombinant protein production in* E. coli. Histidine (His)-tagged p15 and -p16 were amplified using pTRE2pur-p15-Myc and pTRE2pur-p16-Myc as templates, respectively, by PCR using Q5 High-Fidelity DNA Polymerase (New England BioLabs). The primers for His-tagged p15 and His-tagged p16 are listed in Supplementary Table 2. The PCR products were cloned into pET28a (Millipore) and transformed to into BL21(DE3) competent E. coli (New England BioLabs) to produce recombinant His-tagged p15 and His-tagged p16 after the induction by IPTG. The His-tagged proteins were purified on Ni-NTA Superflow Columns (Qiagen).

*p15 N-terminal deletion mutants*. The N-terminal deletion fragments of p15 cDNA were generated by PCR using wild-type p15 cDNA as template with Q5 Site-Directed Mutagenesis Kit (New England BioLabs), with specific primers for Δ2–15, Δ2–27, Δ2–37, and Δ2–49 as listed in Supplementary Table 2.

*Point mutation of p15*. Point mutations were created using site-directed muta-genesis in p15 using two sets of plasmids as templates, i.e., pTRE2pur-p15-myc-tag and pET28a-p15-his-tag, which were used for bladder cancer cell line transfection and E. coli transformation, respectively. The mutagenesis primers for p15/E16A, p15/S20A, p15/R24A, and p15/R45A are listed in Supplementary Table 2.

## Protein–protein interaction and semi-quantification

*Co-immunoprecipitation*. Cultured UMUC3 bladder cancer cells were lysed in RIPA buffer (25 mM Tris-HCl, pH 7.6, 150 mM NaCl, 1% [v/v] NP-40). The soluble proteins were incubated with mouse anti-Myc-tag (Supplementary Table 4). After gentle rocking at 4 °C overnight, agarose-protein A/G beads (sc-2003, Santa Cruz Biotechnology, USA) were then added and incubated at 4 °C for 4 h. The beads were collected by centrifugation, washed, and boiled for 5 min in SDS-PAGE loading buffer. The immunoprecipitated proteins were detected by western blotting using specific antibodies (Supplementary Table 4).

*Pull-down*. Total protein lysates were prepared from cultured UMUC3 cells in RIPA buffer and then incubated with His-tagged proteins produced in E. coli/BL21 with gentle rotation at 4 °C overnight. The mixture was loaded onto Ni-NTA Superflow Columns (Qiagen), incubated and washed, and the bound proteins were eluted with different concentrations of imidazole solution. Western blotting was performed to detect the proteins in each eluted fraction. The bound proteins on western blotting from both immunoprecipitation and pull-down experiments were semi-quantified by scanning the protein bands using NIH ImageJ software.

*Competitive inhibition of CDK binding to p15 by enolase-1*. p15 protein purchased from Abcam was dissolved in 100 μl (2.5 μg/ml) coating buffer (Bio-Rad, Kidlington, UK) and pre-coated onto a clear flat-bottom 96-well plate (Thermo Fisher Scientific) at 4 °C overnight. After washing three times with PBS, the pre-coated wells were blocked with 1% bovine serum albumin (BSA) (W/V) in PBS at 37 °C for 1 h, after which the wells were washed three times with PBS and supplemented with CDK4 (5 μg/ml in 100 μl) (Novus) in the presence of 0–40 μg/ml of enolase-1 (Sino Biological Inc., Wayne, PA). Following additional incubation at 37 °C for 2 h, the wells were washed in PBS containing 0.05% v/v Tween-20 and incubated with 100 μl rabbit anti-CDK4 antibody (Supplementary Table 4) at 37 °C for 1 h. After washing three times in PBS, 100 μl HRP-conjugated goat anti-rabbit IgG antibody (Supplementary Table 4) was added, incubated, and developed in 100 μl of the TMB substrate solution (Thermo Fisher Scientific). Absorbance was read in a microplate reader (Dynex) immediately at the 450 nm after adding 100 μl stop solution (Thermo Fisher Scientific).

*Changes in p15 binding to CDK4 and CDK6 after enolase-1 knockdown*. A Lenti-viral vector (Origene) was reengineered to bear an shRNA of enolase-1 (GCATTGGAGCAGAGGTTTACC-TCAAGAG (loop)-GGCGTTCAATGTCAT CAATGG). The lentivirus was packaged in 293 T cells by co-transfection with the plasmids, VSV-G, PAX2, and the shRNA-bearing vector. The packaged lentivirus was then used to transfect UMUC3 cells harboring Myc-tagged p15, and the stable transfected cells were selected in the presence of 4 μg/ml of blasticidin (InvivoGen). Co-immunoprecipitation of p15 with CDK4 and CDK6 as well as cell cycle and cell proliferation analyses were described above.

## Two-dimensional gel electrophoresis and mass spectrometry of p15-bound proteins. Recombinant proteins (p15-Myc, p16-Myc, and control (myc-tag only)) produced in E. coli that served as baits were separately immobilized on Ni-NTA Superflow Columns and incubated with total proteins extracted from parental UMUC3. After washing, the bound proteins were eluted with imidazole. Two-dimensional electrophoresis was performed based on the carrier ampholyte method of isoelectric focusing[65,66]. Briefly, the first dimension (e.g., isoelectric focusing) was performed using 2% pH 3–10 Isodalt Servalytes (Serva, Heidelberg, Germany). Tropomyosin (MW 33,000 and pI 5.2) served as an internal standard. After equilibration in 10% glycerol, 50 mM dithiothreitol, 2.3% SDS, and 0.0625 M tris, pH 6.8, the first-dimensional gel

was loaded atop a 10% SDS-PAGE, with molecular weight standards (Sigma Chemical Co., St. Louis, MO and EMD Millipore, Billerica MA): myosin (220,000), phosphorylase A (94,000), catalase (60,000), actin (43,000), carbonic anhydrase (29,000), and lysozyme (14,000). Protein spots that were not present in the control or p16-bound fractions were excised, in-gel trypsin-digested, HPLC-purified and subjected to mass spectrometry following established procedures[67].

### Glycolysis analysis

*Enolase activity assay*. Cultured bladder cancer cells ($2 \times 10^6$) were lysed with ice-cold Enolase Assay Buffer (Biovision, Milpitas, CA, USA). The enolase activity and standard curve were both determined according to the instructions of enolase activity colorimetric assay kit (Biovision).

*Seahorse analysis*. Urothelial cells were harvested from the mucosal surfaces of the inside-out mouse bladders. The cells were incubated in a Liberase solution at 37 °C for 15 min, after which the digestion was stopped by adding 2% fetal bovine serum and 0.2 μM EDTA. The cells were then cultured in cell culture microplates pre-coated with poly-L-lysine (Seahorse Bioscience, Santa Clara, CA, USA) at $3 \times 10^4$ cells/well density in 180 μl XF Base DMEM medium (Agilent, Santa Clara, CA, USA) and incubated at 37 °C in 5% $CO_2$. The assays were conducted at different time points by adding glucose, Rot/AA, and 2-DG successively in an XFe24 extracellular flux analyzer (Agilent, Santa Clara, CA, USA), and the ECAR data were recorded automatically.

### Immunofluorescence staining and western blotting.

For staining the de-paraffinized sections were subjected to antigen retrieval by microwaving at the maximum output in antigen unmasking solution (Vector Laboratories, Burlingame, CA, USA) for 20 min. After incubation in 4% Triton X-100 in PBS, the non-specific sites were blocked with 1% BSA in PBS. The primary antibodies used for immuno-fluorescence staining and their dilutions are listed in Supplementary Table 4. After incubation with primary antibody at 4 °C overnight, the sections were washed with PBS three times, and then incubated with Alexa-488-conjugated goat anti-mouse or Alexa-594-conjugated goat anti-rabbit secondary antibody (Invitrogen; 1:500 dilution) at 37 °C for 1 h. After nuclear counterstaining with 1 μg/ml DAPI, the sections were mounted with aqua-mount (Lerner laboratories, Kalamazoo, MI, USA).

For western blotting, cultured human bladder cancer cells or mouse tissues were lysed in whole-cell lysis buffer containing 20 mM Tris, pH 7.6, 50 mM NaCl, 10% SDS, protease, and phosphatase Inhibitors (Thermo Scientific, Rockford, IL, USA). The protein concentrations were determined by Pierce BCA Protein Assay Kit (Thermo Scientific), and 30 μg of the proteins each sample were resolved on SDS-PAGE. The proteins were then electrophoretically transferred onto PVDF membrane, blocked with 5% BSA in TBST, and incubated with the primary antibodies with proper dilutions (Supplementary Table 4). The secondary antibodies used were: horse anti-mouse IgG antibody (1:5000, Cell Signaling Technology), goat anti-rabbit IgG antibody (1:5000, Cell Signaling Technology), and goat anti-chicken (1:3000, Invitrogen). For semi-quantification, the band densities were measured using ImageJ software. Uncropped gels and blots that are the most important to our conclusions are provided in the Source Data as a Source Data file.

### Statistical analysis.

Data were shown as mean ± SD from at least three separate experiments and/or at least three biologically independent samples. Differences between two data sets were determined by Student *t*-tests (two-sided), and differences for the tumor-free rates were analyzed using the Kaplan–Meier method followed by log-rank statistical test using software SPSS v17.0. Differences were considered significant when *P* value is less than 0.05.

### Reporting summary.

Further information on research design is available in the Nature Research Reporting Summary linked to this article.

## Data availability

All the data generated or analyzed during this study are included in this published article and its supplementary information files. The source data on *ENO-1* expression in human bladder cancer cohorts shown in Supplementary Fig. 8 are available in a public repository from the https://www.cbioportal.org/ website. MD simulation data of three protein complexes are available at https://www.nyu.edu/projects/yzhang/MD_data. Source data are provided with this paper.

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

## Acknowledgements

We wish to acknowledge Lijie Ma and Fengxia Zhang for their excellent technical assistance with some of the histopathological analysis. This work was supported in part by National Institutes of Health (P01 CA165980) and awards from the United States Veterans Affairs Office of Research and Development (Biomedical Laboratory Research and Development Service: Merit Review (1I01BX002049) and Research Career Scientist (1IK6BX004479)). Y.Z. wishes to acknowledge the funding from the National Institute of General Medical Sciences (R35 GM127040).

## Author contributions

The first three authors contributed equally to this work. Y.X. designed and carried out the initial in vitro and in vivo experiments, organized the data, and prepared the figures; Y.L. designed and conducted all the in vitro and in vivo experiments during revision; C.Y. and Y.Z. performed the structural analyses and modeling; D.M.S., T.T.S., D.J.D., and M.S.T. contributed to data analysis and interpretation and manuscript editing; X.R.W. conceived the study, helped design the experiments and interpret the data, and wrote and revised the manuscript.

## Competing interests

The authors declare no competing interests.
