## [Peer Review File · Nature Communications]

REVIEWER COMMENTS

Reviewer #1 (Remarks to the Author):

In this manuscript, Xia et al. present compelling evidence for a previously unappreciated role of CDKN2B/p15 as a tumor suppressor in bladder tumorigenesis. This clearly adds to our knowledge on the possible effect of the 9p21.3 loss in bladder cancer, and possibly other cancer types as well. Through genetic and biochemical experiments, Xia et al. show that it is p15 rather than p16 that regulates CDK4/6 activity in urothelial cells, due to stronger protein binding. Also, the authors show p15 to bind and inhibit the activity of Enolase-1, leading to metabolic reprogramming, which is clearly a novel finding.

On the other hand, the mouse phenotype is not striking and it seems that there are some discrepancies with respect both to penetrance and latency. It is to be expected that the combination of a mutant HRas with the loss of a CDK inhibitor could augment the effect of the former. Even if the culprit is p15 and not p16. Besides, the role of Rb proteins in urothelial carcinoma has been addressed in previous papers, by the same group as well. With respect to the action of p15 upon Enolase-1, the work is a bit preliminary. Metabolic processes are quite complex, and the Seahorse analysis alone is not sufficient to link p15-Enolase-1 with metabolic reprogramming. More extensive experimentation including manipulation of Enolase activity in HRas/p16 context would be needed for this. A general comment regarding the manuscript is that image quality is very often poor, which of course could be attributed to image compression during submission. However, in certain H&E images it was impossible to see individual cells and nuclei.

Several comments and suggestions that this reviewer believes would help improve the manuscript are given below:

1. (Ext. Data Fig 1C) Dipstick test revealed differences on the fifth row (arrow) but also on the seventh (moderate) and ninth (intense). Add a comment on what these rows represent.
2. "...revealed markedly enlarged bladders with significantly increased bladder weights in the HRasWT*/p15-/- group, but not in any other groups (Extended Data Fig. 1C, lower panel (B as in bladder)), even though the body weights did not differ significantly (Extended Data Fig. 1D)." change to: "... revealed markedly enlarged bladders (Extended Data Fig. 1C, lower panel (B as in bladder)) with significantly increased bladder weights (Extended Data Fig. 1D, upper panel) in the HRasWT*/p15-/- group, but not in any other groups,), even though the body weights did not differ significantly (Extended Data Fig. 1D, lower panel).
3. "Hydronephrosis was noted only in the HRasWT*/p15-/- group" In Extended Data Fig. 1C, lower panel (K as in kidney), the HRasWT*/p15WT/- is also significantly enlarged as compared to WT and p15WT/-. If it is indeed a representative image then it should be noted accordingly in the text.
4. "particularly in the 6- and 8-month age groups (Extended Data Fig. 1C, lower panel (K as in kidney))," This figure shows 4 months old kidneys. Either rephrase as data not shown (for 6- and 8-month) or adjust further the text accordingly.
5. (Extended Data Fig. 1F) WT H/E is missing. Also, scale bar should be included.
6. (Extended Data Fig. 2) Bad quality images with few exceptions. WT controls are missing, as well as scale bars.
7. Loss of p15INK4B Triggers Early-onset and Highly Penetrant Tumors from Mouse Urothelial Cells Expressing Oncogenic HRas. This section has a very poor description on the histological examination of Ext. Data Fig. 1F samples. Hyperplasias are evident but not described.
8. "oncogenic HRas together are fully tumorigenic in urothelial cells (Extended Data Fig. 1 and" change to: oncogenic HRas together are fully tumorigenic in urothelial cells (Extended

Data Fig. 1E and F and...

9. "(ii) single-allele HRasWT/* mice which we previously showed to consistently develop simple urothelial hyperplasia before 10 months of age;" See comment #8.
10. (Fig 3C) Add in the figure legend that ERK1/2 was also investigated.
11. "In striking contrast and direct support of our initial cohort shown in Extended Data Fig. 1," Rephrase accordingly because the data of Ext Data Fig. 1E and 1D share some striking differences e.g. 20% vs 40% at 2 months or 100% at 8 months vs 80% at 10 months. Here, also, a note on the similar tumorigenic potential of p15WT/- and p16 null mice could be added. The histologic evaluation on figure legend of Fig. 1E should be included in the text (see comment 6).
12. (Extended Data Fig. 3A) Age of samples is not specified. WT or early aged controls are missing. Scale bar is missing.
13. "deletion of both p15 and p16 in HRasWT*/Ink4ab-/- mice did not significantly increase the tumor frequency or shorten the tumor latency (Extended Data Fig. 4)" Tumor frequency is impossible to be deduced from single representative H/E images over time. Sample size is missing. For Ext. Data 4A, the images are of bad quality and also magnification and scale bars are missing. As for the shortened latency, again from the data given it is difficult to be deduced. More time points and increased sample size are needed. Regarding Ext. Data 4B, if the bar for INK4ab-/- represents zero, then a similar one should be added for HRasWT/*. It is not clear what the bar represents. For a cohort of n=6, anything between 0 and 16% does not make sense. In general, this part of the work does not seem robust while the description is not adequate.
14. "loss of both p16 and p15 is not synergistic or significantly additive in urothelial tumor initiation" The presented data are not supportive to such a strong statement.
15. "We found p15 to be significantly more inhibitory of cell proliferation than p16" The expression levels however are not properly calibrated (less p16 is expressed than p51 as Fig 2B indicates).
16. (Extended Data Fig. 5A) Sampling size and timepoint after induction are missing. Also, the G2+M fractions lack statistical significant differences.
17. (Fig 2B) Add the duration of induction prior protein extraction.
18. (Fig 2C) The myc-tag bands in the input WB show the p15 band to run higher than the p16 one. Why is that? Also CDK6 is underexpressed in p15 expressing UMUC3 cells and it contradicts Ext. Data Fig 8C input WB. CDK6 seems rather equally immunoprecipitated by either p15 or p16 in the blot.
19. (Fig 2C-E) The data for p15 binding to CDK4 are consistent (~4 fold more than p16), however, the CDK6 data deviate significantly (2 vs 8 fold between Fig 2C and 2D).
20. (Extended Data Fig. 6A-B) p16 and p16-N/p15-C expressing cells have more phosphorylated pRb1 than p15 and p15-N/p16-C (6A). This is reflected in 6B with higher levels on WST-1 assay. However, p16 and p16-N/p15-C expressing cells also have more phosphorylated pRb1 than controls. This is not reflected in 6B. An explanation for this is needed. Also, it is not clear which groups were compared and what comparison the p values correspond to (p16 and p16-N/p15-C seem to be considered as one group). Sample size and replicates are missing.
21. (Extended Data Fig. 10B) Enolase 1 expression is significantly elevated in p15 null mice. Such a comment should be included in the text.
22. (Fig. 4) Units are missing from A. For B, it would be interesting to see the opposite experiment (immobilization of Enolase 1 and free CDK4) to elaborate on the binding ability of p15 with free vs immobilized substrates. For C, how many shRNAs were tested? The two lanes represent different ones or duplicates of the same? Again n is missing in this figure too.
23. "comprising 8 normal bladder cohorts and 9 bladder tumor cohorts from human patients,"

a reference to cBioportal should be added.

24. "subjected to mass spectrometry following established procedures" A reference is missing.

25. "fluorescence labeled secondary antibody at 37°C for 1 h" Specify which ones and dilution.

26. Materials and Methods should include shRNA experiment and H/E section.

Reviewer #2 (Remarks to the Author):

In this manuscript, Xia and co-workers have combined genetic models of bladder cancer with biochemical and in vitro approaches to compare the tumour suppressive properties of two closely related proteins of the INK4 family, p15 and p16. Both are present in the 9p21.3 chromosomal locus, which is frequently inactivated in cancer. Data presented in this manuscript identify a much more prominent tumour suppressive role of p15 when compared to p16. This solves an existing discrepancy in the field due to the lack of tumorigenicity in vivo previously observed upon combining RAS hyperactivation and p16 loss. Also, the authors have also identified a relevant interaction between enolase-1 of p15 that might connect cell cycle progression and metabolic reprogramming in bladder cancer. Finally, by combining predicted 3D structures, molecular dynamics simulations and in vitro experiments with point mutations the authors have identified the key residues within the N-terminal region that mediate p15 interaction with both CDK4 and enolase-1.

These findings are relevant to the field. In my opinion, the research described here was excellently conceived, executed, interpreted and presented. This reviewer only has some questions to further clarify some of the functional interactions described in the manuscript.

1. Mice combining HRASWT/* and p15wt/- display a lower penetrance and later onset of bladder cancer when compared to the p15 full KO. Have the authors addressed whether the tumours that eventually develop in this p15 heterozygote background display p15 LOH? In other words, is p15 an haplo-insufficient suppressor in this context?

2. Interestingly, the authors have exposed a non-redundant function between p15 and p16 in spite. Interestingly, in vivo experiments (Fig 2) demonstrate that p15 binds CDK4/6 more avidly than p16. Yet, in a p15 deficient background, not only p16 levels increase (Fig 1c) but binding to CDK4 is also enhanced (Ext data Fig 10b). Since p16 cannot bind enolase-1, could the authors comment whether in their opinion the tumour suppressive functions of p15 are mainly related to the metabolic control and not to its role in cell cycle?

3. Similarly, could the authors provide results regarding potential expression changes in p19 expression in the context of p15 or p16 deficient tumours? This is including a p19 western blot in Figure 1c.

4. Finally, several lines of evidence suggest that Rb phosphorylation in S807/S811 might function as CDK4-activated priming sites. CDK4 preferentially phosphorylates S807/S811 in vitro, and deletion of the Cdk4 docking site in Rb abrogates S807/S811 phosphorylation in cells. Yet, the mechanistic implications of the S780 phosphorylation are less clear. It would be interesting to include S807/S811 status in the in vivo experiment shown in Ext data Fig 3c.

Minor points: within the abstract & introduction p16 is referred to as p16INKA when I think it should be p16INK4A

Reviewer #3 (Remarks to the Author):

In the manuscript "Dominant Role of CDKN2B of 9p21.3 Tumor Suppressor Hub in Dual Inhibition of Cell-cycle and Glycolysis," Xia et al demonstrate that p15 potentially plays a more important role than p16 in urothelial cancer progression by more stably interacting with CDK4 and CDK4, and possibly by interacting with enolase1. Overall, these experiments are performed extremely well, with a nice complement of in vivo tumor work and in vitro mechanistic work. The work demonstrating that p15 interacts with CDK4/6 more strongly than p16 is particularly convincing. The data demonstrating that p15 also specifically interacts with enolase 1 is also well done and convincing, and it is very likely that p15 could be affecting glycolytic rates. That said, it remains unclear to what extent the potential activation of glycolysis caused by p15 loss is important for tumorigenesis.

Minor concerns:

1. It would be interesting to know whether HRas p15^{-/-} tumors are sensitive to inhibition of enolase either genetically or with ENOblock.
2. As the authors allude to, ENO1 is often overexpressed in cancers and is associated with grade in bladder cancer. Is it possible that elevated ENO1 could inhibit p15 and promote cell cycle progress in p15 wild type cancer cells?
3. The data in the manuscript would be easier to follow if some of the data in the extended figures is moved to the regular figures. There is a lot of very important data that is easy to miss because it is stuck in the supplement.

Re: Manuscript NCOMMS-20-06853-T, "Dominant Role of CDKN2B of 9p21.3 Tumor Suppressor Hub in Dual Inhibition of Cell-cycle and Glycolysis".

The following is a point-by-point response to the reviewers' critiques and comments (*in italics*) and a summary of our revisions and improvements (regular font).

Reviewer #1 (Remarks to the Author):

In this manuscript, Xia et al. present compelling evidence for a previously unappreciated role of CDKN2B/p15 as a tumor suppressor in bladder tumorigenesis. This clearly adds to our knowledge on the possible effect of the 9p21.3 loss in bladder cancer, and possibly other cancer types as well. Through genetic and biochemical experiments, Xia et al. show that it is p15 rather than p16 that regulates CDK4/6 activity in urothelial cells, due to stronger protein binding. Also, the authors show p15 to bind and inhibit the activity of Enolase-1, leading to metabolic reprogramming, which is clearly a novel finding.

We greatly appreciate these comments.

On the other hand, the mouse phenotype is not striking and it seems that there are some discrepancies with respect both to penetrance and latency. It is to be expected that the combination of a mutant HRas with the loss of a CDK inhibitor could augment the effect of the former. Even if the culprit is p15 and not p16. Besides, the role of Rb proteins in urothelial carcinoma has been addressed in previous papers, by the same group as well. With respect to the action of p15 upon Enolase-1, the work is a bit preliminary. Metabolic processes are quite complex, and the Seahorse analysis alone is not sufficient to link p15-Enolase-1 with metabolic reprogramming. More extensive experimentation including manipulation of Enolase activity in HRas/p16 context would be needed for this. A general comment regarding the manuscript is that image quality is very often poor, which of course could be attributed to image compression during submission. However, in certain H&E images it was impossible to see individual cells and nuclei.

We appreciate the Reviewer's critiques and apologize for the inadequate descriptions in the last version of our manuscript that led the Reviewer into believing that the "*mouse phenotype is not striking*". In fact, a single-copy HRas mutant alone does not induce any urothelial tumor at all before 10 months of age. Thus, that the loss of p15, but not that of p16, in the context of HRas mutation, triggers early-onset (as early as 2 months of age) and highly penetrant urothelial tumors (80% by 4 months in Cohort 1 (Fig. 1E) and 70% by 6 months in Cohort 2 (Fig. 2D)) is a very striking finding. Although there are indeed some differences in tumor onset and penetrance between these two cohorts, both cohorts clearly show the strong cooperativity between p15 loss and HRas mutation. The Reviewer correctly noted the low frequency of tumors also present in mice with p16 loss and HRas mutation. However, not only are the tumor frequencies (only 10% between 6-8 months), but also the onset and tumor burden, in these mice are considerably lower than mice with p15 loss and HRas mutation. As we described previously, tumors in p16^{-/-}/HRas mice were very small in size and could only be detected microscopically. In response to the Reviewer's valid criticism, we have provided much clearer descriptions in the revised text by: (i) emphasizing that HRas mutant alone is not tumorigenic; (ii) citing our previous study showing the complete lack of cooperativity between HRas mutation and loss of even both p16 and p19 (Mo L., et al., J.

Clin. Invest. 2007; 117:314-325); (iii) drawing additional attention to the fact that HRas mutation and p15 loss are highly cooperative, inducing early-onset and much more penetrant tumors that are highly conspicuous by gross anatomy than HRas mutation and p16 loss; (iv) emphasizing that the differences between p15^{-/-}/HRas mice and p16^{-/-}/HRas mice reside with not only tumor frequency, but also tumor onset and tumor burden; and (v) acknowledging the differences between the two p15^{-/-} cohorts and discussing the potential underlying cause in terms of slight genetic background differences (although we had adhered to the industry-standard by back-crossing for at least 6 generations from the C57BL/6 into the FVB/N background, the cohort in Fig. 1E was closer to the C57BL/6 background and the cohort Fig. 2D was closer to FVB/N). We have endeavored to clarify these points in our revised manuscript.

In addition, we were able during the revision process to breed and analyze a third cohort comparing HRas-mediated urothelial tumorigenesis in the context of p15 deficiency versus p16 deficiency. Because of the severe constraints during the ongoing Covid-19 pandemic and because it is significantly more meaningful to assess tumorigenesis during the early ages (as late-stage tumorigenesis is known to be more associated with stochastic events), we analyzed mice at 3 months of age. We found urothelial tumors in 40% of the HRas/p15^{-/-} mice and 0% of the HRas/p16^{-/-} mice (new Supplementary Fig. 2). We have therefore demonstrated the early-onset urothelial tumorigenesis in HRas/p15^{-/-} mice in three independent cohorts (Fig. 1E, Fig. 2D and Supplementary Fig. 2A). We have also demonstrated, in striking contrast, the insufficient cooperativity between HRas and p16 deficiency in three different cohorts (Fig. 2D, Supplementary Fig. 2 and Mo et al, JCI, 117:314-325, 2007). Our data on the role of p15 versus p16 in urothelial tumor suppression are therefore quite reproducible. In response to the Reviewer's constructive comment about the differences in tumor penetrance in HRas/p15^{-/-} mice between different mouse cohorts, we have changed our previous descriptive term on tumor penetrance from "fully penetrant" to "highly penetrant".

With respect to Rb proteins in urothelial carcinoma, our group indeed previously studied the issue, but it was strictly in the context of their relationship with p53 in the initiation of high-grade, invasive bladder tumors. In that study, we concluded that the inactivation of the entire Rb family (Rb1, p107 and p130) is required for the initiation of muscle-invasive bladder cancer (He F et al., Cancer Res. 2009; 69:9413-9421). That paper was distinctly from our current paper dealing with the role and mechanisms of action of p15 in the genesis of low-grade, non-invasive bladder tumors. Here the phosphorylation status of Rb proteins serves as functional readout for p15, p16, CDK4 and CDK6.

Regarding p15, enolase-1, metabolic reprogramming and tumorigenesis, we did not intend to address the role of enolase-1 in glycolysis or the role of glycolysis in tumorigenesis, both of which have been well established in the literature (Jung et al., Biochem. Soc. Trans., 42:1756-61, 2014; Baig et al., Semin. Cancer Biol. 56:1-11, 2019; Yang et al., Cell Death Dis., 11:870, 2020). Instead, our finding centers on linking p15 with enolase-1, which is entirely new. Our suggestion that p15 may play a role in binding and inhibiting enolase-1 is based on multiple lines of evidence: (i) p15, but not p16, "fishes out" enolase-1 from total protein extracts of UMUC3 cells in unbiased proteomic screening assays (Supplementary Fig. 7; Supplementary Table 1); (ii) native p15, but not native p16, co-immunoprecipitates with enolase-1 from total protein extracts of UMUC3 cells (Fig. 7A, right column); (iii) bacteria-produced recombinant p15, but not bacteria-produced recombinant p16, pulls down enolase-1 from total protein extracts of UMUC3 cells (Fig. 7A, left column); (iv) native p15, but not native p16, co-immunoprecipitates with

enolase-1 from in vivo transgenic mouse urothelial cells expressing oncogenic HRas (Fig. 7B); (v) like the binding to CDK4, it is the N-terminus of p15 that binds to enolase-1, thus suggesting a competitive binding relationship among the three proteins (Fig. 7C; and see below); (vi) purified p15, but not purified p16, markedly inhibits the activity of purified enolase-1 in the test tubes (Fig. 8A); (vii) purified enolase-1 competes the binding between p15 and CDK4 in a saturable manner; and (viii) markedly increased glycolysis, glycolytic capacity and glycolytic reserve are all significantly associated with urothelial cells expressing HRas and lacking p15, compared to urothelial cells expressing HRas alone (Fig. 8D), as evidenced by Seahorse analysis, which is presently an industry-standard in assessing glycolytic status. Taken together, we believe our data demonstrating the interaction between p15 and enolase-1 are quite strong. As mentioned, the role of enolase-1 and glycolysis in tumorigenesis has been well established by other investigators. While in vivo evidence, such as urothelium-specific knockout or chemical inhibition of enolase-1, is necessary to prove the causative relationship among p15, enolase-1 and tumorigenesis, the evidence we have provided here, particularly the fact that p15 binds and inhibits enolase-1 activity and competes binding with CDK4, has set the stage for additional systematic in vitro and in vivo studies to be carried out by us or others in the future.

To move toward this direction during our manuscript revision process, we have performed functional experiments to assess the effects of down-regulating enolase-1 on cell proliferation, making use of cultured bladder cancer cell lines we had engineered. As presented in Fig. 9A, down-regulation of enolase-1 significantly reduces cell-cycle progression (Fig. 9B) and proliferation (Fig. 9C). This was associated with an increased binding between p15 and CDK4/6 and reduced phosphorylation of pRB (Fig. 9A), again supporting the interrelationship among enolase-1, p15 and CDK4/6.

About the low quality of some of the pdf files provided for the last review, we sincerely apologize, and we have endeavored to make the necessary improvements during our revision. Please be assured that high-quality images will be provided later on as per Journal requirement.

1. (Ext. Data Fig 1C) Dipstick test revealed differences on the fifth row (arrow) but also on the seventh (moderate) and ninth (intense). Add a comment on what these rows represent.

Due to very limited amounts of mouse urine, we were only able to apply the samples to the one row for hematuria test. We have now eliminated the other rows to avoid confusion.

2. “revealed markedly enlarged bladders with significantly increased bladder weights in the HRasWT/p15-/- group, but not in any other groups (Extended Data Fig. 1C, lower panel (B as in bladder)), even though the body weights did not differ significantly (Extended Data Fig. 1D).” change to: “ revealed markedly enlarged bladders (Extended Data Fig. 1C, lower panel (B as in bladder)) with significantly increased bladder weights (Extended Data Fig. 1D, upper panel) in the HRasWT*/p15-/- group, but not in any other groups,), even though the body weights did not differ significantly (Extended Data Fig. 1D, lower panel).*

Changes have been made as suggested.

3. *“Hydronephrosis was noted only in the HRasWT*/p15-/- group” In Extended Data Fig. 1C, lower panel (K as in kidney), the HRasWT*/p15WT/- is also significantly enlarged as compared to WT and p15WT/-. If it is indeed a representative image then it should be noted accordingly in the text.*

While the kidney shown under HRasWT*/p15WT/- was slightly bigger, there was clearly no hydronephrosis in this kidney – a finding confirmed after the kidney was sliced open across the renal pelvis. We have made a note of it in the revised text.

4. *“particularly in the 6- and 8-month age groups (Extended Data Fig. 1C, lower panel (K as in kidney)),” This figure shows 4 months old kidneys. Either rephrase as data not shown (for 6- and 8-month) or adjust further the text accordingly.*

The previous statement in the text has been corrected to indicate 4-month old mice.

5. *(Extended Data Fig. 1F) WT H/E is missing. Also, scale bar should be included.*

The WT H/E images and the scale bars have been added.

6. *(Extended Data Fig. 2) Bad quality images with few exceptions. WT controls are missing, as well as scale bars.*

We apologize for the poor quality of the images, and we have now included higher quality images. Scale bar has been added as well.

7. *Loss of p15INK4B Triggers Early-onset and Highly Penetrant Tumors from Mouse Urothelial Cells Expressing Oncogenic HRas. This section has a very poor description on the histological examination of Ext. Data Fig. 1F samples. Hyperplasias are evident but not described.*

We appreciate the Reviewer’s critique, and have improved the description in the revised text.

8. *“oncogenic HRas together are fully tumorigenic in urothelial cells (Extended Data Fig. 1 and” change to: oncogenic HRas together are fully tumorigenic in urothelial cells (Extended Data Fig. 1E and F and*

The changes have been made as suggested.

9. *“(ii) single-allele HRasWT*/ mice which we previously showed to consistently develop simple urothelial hyperplasia before 10 months of age;” See comment #8.*

The changes have been made as suggested.

10. *(Fig 3C) Add in the figure legend that ERK1/2 was also investigated.*

The Reviewer most likely meant Fig. 1C (now Fig. 2C) as there was no ERK1/2 in the old Fig. 3C. We have made the change Fig. 2C accordingly. Additional information on p19 was added as per Reviewer 2's recommendation.

11. *"In striking contrast and direct support of our initial cohort shown in Extended Data Fig. 1," Rephrase accordingly because the data of Ext Data Fig. 1E and 1D share some striking differences e.g. 20% vs 40% at 2 months or 100% at 8 months vs 80% at 10 months. Here, also, a note on the similar tumorigenic potential of p15^{WT/-} and p16 null mice could be added. The histologic evaluation on figure legend of Fig. 1E should be included in the text (see comment 6).*

Changes made as suggested.

12. *(Extended Data Fig. 3A) Age of samples is not specified. WT or early aged controls are missing. Scale bar is missing.*

Changes made as suggested.

13. *"deletion of both p15 and p16 in HRas^{WT/*}/Ink4ab^{-/-} mice did not significantly increase the tumor frequency or shorten the tumor latency (Extended Data Fig. 4)" Tumor frequency is impossible to be deduced from single representative H/E images over time. Sample size is missing. For Ext. Data 4A, the images are of bad quality and also magnification and scale bars are missing. As for the shortened latency, again from the data given it is difficult to be deduced. More time points and increased sample size are needed. Regarding Ext. Data 4B, if the bar for INK4ab^{-/-} represents zero, then a similar one should be added for HRas^{WT/*}. It is not clear what the bar represents. For a cohort of n=6, anything between 0 and 16% does not make sense. In general, this part of the work does not seem robust while the description is not adequate.*

These are all valid critiques to which we have addressed accordingly. We generated another cohort comprising the following genotypes: Ink4ab^{-/-}, HRas^{WT/*}, HRas^{WT/*}/Ink4ab^{-/-} and HRas^{WT/*}/p15^{-/-}. Upon analysis of 4 months old mice (see aforementioned reasoning in analyzing younger mice), we observed bladder tumor formation in 60% of the HRas^{WT/*}/Ink4ab^{-/-} mice and 53.3% of the HRas^{WT/*}/p15^{-/-} mice (Supplementary Fig. 3). While the tumor rate is slightly higher in HRas^{WT/*}/Ink4ab^{-/-} mice than in HRas^{WT/*}/p15^{-/-}, there is virtually no statistical significance. These results are therefore consistent with our previous cohort and further support our suggestion that deletion of both p15 and p16 in the context of HRas* did not significantly increase the tumor frequency or shorten the tumor latency behind deletion of p15 alone. We have also included more images from both cohorts (Supplementary Fig. 3B and 3C). We have indicated the sample sizes in the figure legend, improved the quality of the images, added scale bars, corrected mistake in the bar diagram for the previous cohort and improved the description in the text.

14. *"loss of both p16 and p15 is not synergistic or significantly additive in urothelial tumor initiation" The presented data are not supportive to such a strong statement.*

This is the critique we have addressed in the last paragraph. Additionally, we have softened our statement to say that it appears from the data we presented here that the loss of both p16 and p15 is not significantly

synergistic in urothelial tumor initiation, although further investigation with large cohorts and more time points is necessary to firmly establish this conclusion.

15. “We found p15 to be significantly more inhibitory of cell proliferation than p16” The expression levels however are not properly calibrated (less p16 is expressed than p51 as Fig 2B indicates).

As we previously stated, the cell proliferation, shown in Fig. 2A (now Fig. 4A), had already been calibrated (normalized) in reference to the expression levels of p15 and p16, shown in Fig. 2B (now Fig. 4B). In other words, the proliferation-inhibitory effects of p15 versus p16 had already taken into account the slightly different levels of these two proteins. The fact that we employed the identically tagged p15 and p16 was also intended to minimize the potential differences during normalization using anti-p15 and anti-p16 antibodies with different affinities.

16. (Extended Data Fig. 5A) Sampling size and timepoint after induction are missing. Also, the G2+M fractions lack statistical significant differences.

We apologize for the inadvertent omissions, and have made the corrections in the revised figure.

17. (Fig 2B) Add the duration of induction prior protein extraction.

This has been added.

18. (Fig 2C) The myc-tag bands in the input WB show the p15 band to run higher than the p16 one. Why is that? Also CDK6 is underexpressed in p15 expressing UMUC3 cells and it contradicts Ext. Data Fig 8C input WB. CDK6 seems rather equally immunoprecipitated by either p15 or p16 in the blot.

The differences between p15 and p16 in binding to CDK6 were normalized twice in reference to immunoprecipitated, myc-tagged p15 or p16 and then to CDK4 or CDK6. While the differences were clear (old Fig. 2C, lower panels), we agree with reviewer that it was less than striking for this important experiment. In response to his/her valid concern, we have repeated the experiment (new Fig. 4C) by better normalizing the input. The new data are much more compelling in demonstrating p15 as a stronger binder to both CDK4 and CDK6 than p16 (also see below).

19. (Fig 2C-E) The data for p15 binding to CDK4 are consistent (~4 fold more than p16), however, the CDK6 data deviate significantly (2 vs 8 fold between Fig 2C and 2D).

As mentioned in the last point, we repeated the experiments shown in Fig. 2C (now Fig. 4C), and our new data shown in the new Fig. 4C are highly consistent with our previous results and further demonstrate the stronger binding between p15 and CDK4/CDK6 than p16. Fig. 2C (now Fig. 4C) is from the immunoprecipitation experiments, which reflects CDK4/CDK6 that naturally bind to p15, whereas Fig. 2D and 2E (now Fig. 4D and Fig. 4E) are from the pull-down experiments using bacteria-produced, experimentally immobilized, recombinant p15 or p16 to pull down the free CDK4/CDK6 in total protein extracts. While we do not expect the results from these two different approaches to be entirely the same,

the results from both approaches clearly show much stronger binding between p15 and CDK4/CDK6 than between p16 and CDK4/CDK6. We have endeavored to make these points clearer in our revised text.

20. *(Extended Data Fig. 6A-B) p16 and p16-N/p15-C expressing cells have more phosphorylated pRb1 than p15 and p15-N/p16-C (6A). This is reflected in 6B with higher levels on WST-1 assay. However, p16 and p16-N/p15-C expressing cells also have more phosphorylated pRb1 than controls. This is not reflected in 6B. An explanation for this is needed. Also, it is not clear which groups were compared and what comparison the p values correspond to (p16 and p16-N/p15-C seem to be considered as one group). Sample size and replicates are missing.*

We wish to clarify that we generally assess pRb1 phosphorylation at earlier time points than cell proliferation, because changes in the former take place immediately (and level off gradually) after the induced expression of p15, p16 or their mutants, and changes in the latter are time-dependent, as evidenced in Extended Data Fig. 6B (now Fig. 5B). Western blotting shown in Extended Data Fig. 6B (now Fig. 5B) was done 24 hours after doxycycline induction, at which point differences in cell proliferation were not apparent. Regarding comparisons between the groups, p16 and p16-N/p15-C were considered as one group because of their very close values to compare individually with p15 or p15-N/p16-C. These issues have been clarified in the legend and text.

21. *(Extended Data Fig. 10B) Enolase 1 expression is significantly elevated in p15 null mice. Such a comment should be included in the text.*

We have added a comment on this.

22. *(Fig. 4) Units are missing from A. For B, it would be interesting to see the opposite experiment (immobilization of Enolase 1 and free CDK4) to elaborate on the binding ability of p15 with free vs immobilized substrates. For C, how many shRNAs were tested? The two lanes represent different ones or duplicates of the same? Again n is missing in this figure too.*

Units (optical density, O.D.) are now added in the Y axis label. For B, we chose the increasing amounts of enolase-1 because enolase-1 is over-expressed in human tumors, which may serve a competitive inhibitor to disable p15 when p15 locus is wild-type or heterozygous. This has been additionally discussed as per Reviewer 3's suggestion. For C (now Fig. 9A), two shRNAs were tested with (consistent) results with one shown. The two lanes were from the two stably transfected clones. These have been explained in the text, with additional data provided from Fig. 9B through 9C.

23. *“comprising 8 normal bladder cohorts and 9 bladder tumor cohorts from human patients,” a reference to cBioportal should be added.*

This is a website, a link to which has now been provided.

24. *“subjected to mass spectrometry following established procedures” A reference is missing.*

Reference added.

25. “fluorescence labeled secondary antibody at 37???” for 1 h” Specify which ones and dilution.

Information provided.

26. *Materials and Methods* should include shRNA experiment and H&E section.

We have added the description on shRNA knockdown of enolase-1 and H&E section processing in the revised *Materials and Methods*.

Reviewer #2 (Remarks to the Author):

In this manuscript, Xia and co-workers have combined genetic models of bladder cancer with biochemical and in vitro approaches to compare the tumour suppressive properties of two closely related proteins of the INK4 family, p15 and p16. Both are present in the 9p21.3 chromosomal locus, which is frequently inactivated in cancer. Data presented in this manuscript identify a much more prominent tumour suppressive role of p15 when compared to p16. This solves an existing discrepancy in the field due to the lack of tumorigenicity in vivo previously observed upon combining RAS hyperactivation and p16 loss. Also, the authors have also identified a relevant interaction between enolase-1 of p15 that might connect cell cycle progression and metabolic reprogramming in bladder cancer. Finally, by combining predicted 3D structures, molecular dynamics simulations and in vitro experiments with point mutations the authors have identified the key residues within the N-terminal region that mediate p15 interaction with both CDK4 and enolase-1. These findings are relevant to the field. In my opinion, the research described here was excellently conceived, executed, interpreted and presented.

We are very grateful for these remarks.

This reviewer only has some questions to further clarify some of the functional interactions described in the manuscript.

1. Mice combining HRAS^{WT/} and p15^{wt/-} display a lower penetrance and later onset of bladder cancer when compared to the p15 full KO. Have the authors addressed whether the tumours that eventually develop in this p15 heterozygote background display p15 LOH? In other words, is p15 an haplo-insufficient suppressor in this context?*

We very much appreciate the Reviewer’s insightful comments. We actually did assess using Real-time quantitative PCR the status of the second p15 allele in the relatively small number and microscopic level of the tumors in the HRAS^{WT/*}/p15^{WT/-} mice. While the sample size (n=3) was too small to report, we did not see the structural loss of the second allele. Nonetheless, we cannot rule out the possibility that the remaining p15 could be inactivated by other mechanisms, such as hypermethylation, antisense long non-coding RNA ANRIL at the 9p21.3 locus, deficiency of TGF-beta signaling which normally upregulates p15 or overexpression of enolase-1 which could functionally disable p15 (see later). This is such an important issue, given the prevalence of heterozygous deletion of 9p21.3 locus in human tumors, that it

warrants a dedicated study in mouse as well as human settings in the future. In response to the Reviewer's comments, we have added a few sentences in the Discussion to address this issue.

2. Interestingly, the authors have exposed a non-redundant function between p15 and p16 in spite. Interestingly, in vivo experiments (Fig 2) demonstrate that p15 binds CDK4/6 more avidly than p16. Yet, in a p15 deficient background, not only p16 levels increase (Fig 1c) but binding to CDK4 is also enhanced (Ext data Fig 10b). Since p16 cannot bind enolase-1, could the authors comment whether in their opinion the tumour suppressive functions of p15 are mainly related to the metabolic control and not to its role in cell cycle?

The Reviewer raised an interesting point. We believe that the tumor suppressive activities of p15 on cell-cycle progression vis-à-vis metabolic control depend on several different scenarios. Firstly, under normal condition where both p15 and p16 are present and the level enolase-1 is low level (i.e., low aerobic glycolysis), p15 binds CDK4/6 much more strongly than p16, as we demonstrated here with in vivo, cell culture and structural studies. As a result, p15 acts as a potent cell-cycle inhibitor, without having much effect on glycolysis which is not essential in normal cells. Secondly, when p15 is lost in tumor cells, more p16 binds CDK4/6, because the stronger, competitive binding from p15 is absent. Since p16 is a weaker binder/inhibitor of CDK4/6, the compensatory induction of p16 is insufficient to block HRas-mediated tumorigenesis. Also, p16 does not bind or inhibit enolase-1, and enolase-1 is unopposed functionally, thus strongly promoting glycolysis. Thirdly, as Reviewer 3 pointed out, in some tumors where p15 is wild-type or heterozygous, and enolase-1 is highly upregulated, enolase-1 may exert dual tumorigenic effects by binding and functionally inactivating p15 as well as by promoting glycolysis. Finally, we inhibited enolase-1 by shRNA knockdown in UMUC cells (Fig. 9A-C) and observed significant inhibition of cell-cycle progression and proliferation. Thus, based on the fact that p15 binds and inhibits both CDK4/CDK6 and enolase-1 in vitro and in vivo, we suggest that p15 exerts dual inhibitory effects on cell-cycle progression and glycolysis. We have endeavored to discuss these scenarios more clearly in our revised manuscript.

3. Similarly, could the authors provide results regarding potential expression changes in p19 expression in the context of p15 or p16 deficient tumours? This is including a p19 western blot in Figure 1c.

We repeated the entire experiment by adding an anti-p19 antibody and included the data in the new Fig. 2C. Interestingly, p19 is markedly upregulated almost exclusively in urothelial cells of the HRas^{WT}*/p15^{-/-} mice. Given the fact that p19 is a positive regulator of p53, this could be interpreted as a compensatory tumor defense against progression from low-grade non-invasive to the high-grade invasive stage. It would be highly worthwhile in the future for us or other investigators to study the effects of loss of p15 and p19 or that of p15, p16 and p19 altogether in urothelial tumor formation and progression.

4. Finally, several lines of evidence suggest that Rb phosphorylation in S807/S811 might function as CDK4-activated priming sites. CDK4 preferentially phosphorylates S807/S811 in vitro, and deletion of the Cdk4 docking site in Rb abrogates S807/S811 phosphorylation in cells. Yet, the mechanistic implications of the S780 phosphorylation are less clear. It would be interesting to include S807/S811 status in the in vivo experiment shown in Ext data Fig 3c.

In response to the Reviewer's suggestion, we have included Western blotting using an anti-S807/S811 of Rb. The result shows that S807/S81 was indeed hyper-phosphorylated in the absence of p15 (new Fig. 3C). This is in keeping with our other data demonstrating that p15 strongly binds with and inhibits CDK4 (Fig. 4C and 4D; Fig. 5C and 5D; Fig. 6C and 6D), and that restoration of p15 expression markedly reduces S807/S81 phosphorylation (Fig. 4B and Fig. 5A).

Minor points: within the abstract & introduction p16 is referred to as p16INKA when I think it should be p16INK4A

Response: Changes made as pointed out correctly by the Reviewer.

Reviewer #3 (Remarks to the Author):

In the manuscript "Dominant Role of CDKN2B of 9p21.3 Tumor Suppressor Hub in Dual Inhibition of Cell-cycle and Glycolysis", Xia et al demonstrate that p15 potentially plays a more important role than p16 in urothelial cancer progression by more stably interacting with CDK4 and CDK4, and possibly by interacting with enolase1. Overall, these experiments are performed extremely well, with a nice complement of in vivo tumor work and in vitro mechanistic work. The work demonstrating that p15 interacts with CDK4/6 more strongly than p16 is particularly convincing. The data demonstrating that p15 also specifically interacts with enolase 1 is also well done and convincing, and it is very likely that p15 could be affecting glycolytic rates.

We greatly appreciate these comments.

That said, it remains unclear to what extent the potential activation of glycolysis caused by p15 loss is important for tumorigenesis.

Minor concerns:

1. It would be interesting to know whether HRas p15^{-/-} tumors are sensitive to inhibition of enolase either genetically or with ENOblock.

This is indeed an interesting study worthy of pursuing in a systematic manner. At the present time, however, we do not have sufficient number of animals that are required (in the order of hundreds of mice and spanning many months) to tease out the formulation, doses, dosing schedule, delivery route, bioavailability, efficacy and toxicity of ENOblock in our models. We were not allowed to breed new animals during the Covid-19 lockdown – a restriction that still has not been fully lifted at the time of preparing this response. Nonetheless, in response to the Reviewer's valid point, we have tested the effects of shRNA knockdown of enolase-1 on cultured UMUC3 cells (Fig. 9A). We did see an appreciable reduction of cell-cycle progression and cell proliferation (new Fig. 9B and 9C). In conjunction with the fact that knock-down of enolase-1 increased the binding between p15 and CDK4/CDK6 and reduced Rb phosphorylation (Fig. 9A), inhibiting enolase-1 may well have the potential of restoring the cell-cycle inhibitory function of p15 (also see below).

2. As the authors allude to, ENO1 is often overexpressed in cancers and is associated with grade in bladder cancer. Is it possible that elevated ENO1 could inhibit p15 and promote cell cycle progress in p15 wild type cancer cells?

We agree with the Reviewer completely in that elevated enolase-1, often observed in tumor cells, could bind and functionally disable the wild-type p15 (or the heterozygous p15 as Reviewer 2 alluded to), thus promoting cell-cycle progression. Consistent with this notion, we showed in Fig. 9A that the knockdown of enolase-1 significantly increased the binding of p15 to CDK4 and CDK6 (as evidenced by co-immunoprecipitation) and decreased Rb phosphorylation, likely as a result of the reduced binding between enolase-1 and p15. This adds another mechanism as to how enolase-1 can indirectly affect cell-cycle progression and proliferation. In response to the Reviewer's valid point, we have expanded our discussion about this in our revised text.

3. The data in the manuscript would be easier to follow if some of the data in the extended figures is moved to the regular figures. There is a lot of very important data that is easy to miss because it is stuck in the supplement.

Our manuscript was initially submitted to another journal which allows fewer display items in the body of the paper. The Reviewer's point is well taken, and we are now including 9 regular figures in the main body and 10 other figures and 4 tables as Supplementary Materials.

REVIEWER COMMENTS

Reviewer #1 (Remarks to the Author):

The authors have addressed all my concerns in the revised manuscript not only editorially but more importantly by performing new experiments including a new cohort of mice.

Reviewer #2 (Remarks to the Author):

The authors have now convincingly addressed all my concerns.

Reviewer #3 (Remarks to the Author):

The authors have addressed all of my concerns.

Reviewer #4 (Remarks to the Author):

I have evaluated in more detail the technical aspects related to presented molecular dynamics simulation aspect of this work. In the manuscript, the authors utilize classical MD simulations to find further insights and rationale into the observed interaction differences on the molecular level between p15 and p16 for CDK4/CDK6. Although the basis for the simulations seems ok, I have some serious concerns of the authors interpretation of the data at hand.

Major issues

There are clearly major caveats understanding what can be interpreted from the timescale of the simulations, especially when combined with the low number of replicas. First, the simulations timescale is insufficient for a proper decent side chain sampling [1]. Usually, microsecond timescale is required for this. Second, only one or two replicas are simulated for each studied system. This provides a high-risk for author biased false positive conclusions [2]. On top of this, I am extremely worried here, as the starting configurations of the simulations are not based on high-quality experimental data, but low-quality structures combined with homology models. This will provide high uncertainty for the conducted simulations and is especially risky when combined with short simulations.

1 Henzler-Wildman K and Kern D: Dynamic personalities of proteins. *Nature*. 2007, 450: 964–972

2 Avoiding False Positive Conclusions in Molecular Simulation: The Importance of Replicas. *J. Chem. Theory Comput.* 2018, 14, 12, 6127–6138

Overall, the authors make too definitive statements of the interactions based on the highly uncertain, most probably under sampled data. The presented data is suggestive at most; therefore, statements in the paper based on this data such as: “Our modeling therefore provides a structural basis extending the experimental results indicating that p15 interacts much more strongly with CDK6 than p16.” are clearly overstatements. To make such statements, more comprehensive MD simulations (more replicas, and starting from different conformations, see below especially related to interactions; and preferably longer

simulations) are required to ensure that the data is valid to justify these.

Detailed concerns

[1] According to Figure 6A, in the models in the (starting?) conformation of the p16, R22, E9 and E12 are located far from each other. Based on this figure, similar conformation as with p15 would be possible just with slight side chain arrangement. Therefore, I am worried if the observed salt bridge / H-bond differences arise just based on the starting conformation of the simulations? This is not a minor point as this is used as a major explanation in the manuscript. This could be easily demonstrated by having good coverage of replicas starting for both conformations for p15 and p16. If these interactions are the definitive events, it should be seen in all simulations how p15 obtains the interactions regardless of the starting conformation and p16 loses (or do not obtain) these interactions with different starting conformations.

[2] The authors state that improved binding of p16(R24L) arises from the repulsion of R24 from this position. This is unclear if it would arise from repulsion of R24 or actually improved affinity arising from L24 (burying lipophilic residue from the solvent in the protein-protein interface usually increases binding affinity). Are there lipophilic contacts available for L24? The observed interactions in the close proximity of the position 24, (for both R24 or L24), needs to be qualified and quantified. Also, what is the role for the starting conformation for the observed results as such short simulation was used?

[3] Amino acids Glu9 and Glu12 of CDK6 are displayed to interact with p15. However, based on the CDK6 sequence (1-MEKDGLCRADQQ-12...) the residues in these positions are Ala9 and Gln12. Please assign the correct residue numbers.

[4] The authors demonstrate the different deletions of N-terminal part reduces/abolishes binding to CDK4. However, all molecular modeling explanations are based on CDK6. How does this fit to your model? Although they are highly similar, please, demonstrate all similarities and dissimilarities of CDK4 and CDK6 on the putative p15/p16 interaction interfaces.

[5] The authors are referring to the wrong supplementary fig. in: "We identified several key interactive points between the N-terminus of p15 and CDK6, particularly in the salt-bridges and H-bonds (Fig. 6A; Supplementary Fig. 5A-D)"

[6] In supplementary figures, are the shown H-bond probability towards side chain, backbone or both? This is crucial as the authors are stating is as a salt bridge. Please differentiate clearly backbone H-bonds and salt bridges in the text and in the figures.

[7] In the methods, a clustering approach was used for "binding interface analysis". First, I am puzzled why the authors applied such a method for these short trajectories? Furthermore, it is unclear in the manuscript where this method was applied, please clarify. I suspect that this type of approach will only further bias already biased (under sampled) data.

[8] The overall stability and/or instability of the interacting protein, needs to be quantitatively described. For this, e.g. RMSF values would be useful.

[9] The mouse and human p15 are not totally identical (shorter N-terminal in mouse and some amino acid differences). How does this reflect to your modelling explanation of the

observed interactions?

[10] In supplementary figure 10 RMSD values of systems are presented. Which RMSD is this, protein backbone or does it include also sidechains or what? Please define.

[11] Describe how replicas differ from each other, according to the system setup the initial configuration was identical. I was unable to locate this information from the manuscript.

[12] I wonder why such a small timesteps as 5 ps were used for the analysis? "The atomic coordinates of the complexes were saved every 5 ps to obtain the trajectories for post structural analysis"

[13] The authors have not uploaded the raw simulation data readily available to one of the public repositories. This is highly recommended practice in the field and provides further confidence for the simulation data³.

3 Abraham M et al.: Sharing Data from Molecular Simulations. *J Chem Inf Mod* 2019, 59, 10, 4093–4099

Reviewer # 4 Comments

I have evaluated in more detail the technical aspects related to presented molecular dynamics simulation aspect of this work. In the manuscript, the authors utilize classical MD simulations to find further insights and rationale into the observed interaction differences on the molecular level between p15 and p16 for CDK4/CDK6. Although the basis for the simulations seems ok, I have some serious concerns of the authors interpretation of the data at hand.

Major issues

There are clearly major caveats understanding what can be interpreted from the timescale of the simulations, especially when combined with the low number of replicas. First, the simulations timescale is insufficient for a proper decent side chain sampling [1]. Usually, microsecond timescale is required for this. Second, only one or two replicas are simulated for each studied system. This provides a high-risk for author biased false positive conclusions [2]. On top of this, I am extremely worried here, as the starting configurations of the simulations are not based on high-quality experimental data, but low-quality structures combined with homology models. This will provide high uncertainty for the conducted simulations and is especially risky when combined with short simulations.

1 Henzler-Wildman K and Kern D: Dynamic personalities of proteins. Nature. 2007, 450: 964–972

2 Avoiding False Positive Conclusions in Molecular Simulation: The Importance of Replicas. J. Chem. Theory Comput. 2018, 14, 12, 6127–6138

Overall, the authors make too definitive statements of the interactions based on the highly uncertain, most probably under sampled data. The presented data is suggestive at most; therefore, statements in the paper based on this data such as: “Our modeling therefore provides a structural basis extending the experimental results indicating that p15 interacts much more strongly with CDK6 than p16.” are clearly overstatements. To make such statements, more comprehensive MD simulations (more replicas, and starting from different conformations, see below especially related to interactions; and preferably longer simulations) are required to ensure that the data is valid to justify these.

Response: Thanks for the reviewer’s careful reading and detailed comments. We agree that molecular modeling and molecular dynamics simulation results are suggestive in nature, and in our revised manuscript, first we changed these definitive statements to be more suggestive; second, following the reviewer’s suggestion, we added more replicas regarding to CDK6-p15 and CDK6-p16 and CDK6-16(R24L)

models. For each model, we added up to 5 replicas, and each replica's timescale is 150-ns (see revised Supplemental Table 3). Our revised MD results are based on these updated MD replicas, and are consistent with our previous findings.

On the other hand, we respectfully disagree with the statement "First, the simulations timescale is insufficient for a proper decent side chain sampling [1]. Usually, microsecond timescale is required for this." Actually from the reference 1, it is stated that: "We distinguish between tier-1 and tier-2 substates as small groups of atoms fluctuating collectively on the nanosecond timescale (such as loop motions) and local atomic fluctuations on the picosecond timescale (such as side-chain rotations), respectively (Fig. 1)." Since the side-chain rotation motion is typical in picosecond timescale, our simulation length (150 ns for each replica) is reasonable.

Supplemental Table 3

System	Atoms	Box size	Ions	Time for each replica	Num of replicas
p16-CDK6 complex	71,906	85 × 98 × 103	2 Na ⁺	150 ns	5
p15-CDK6 complex	66,357	86 × 88 × 105	2 Na ⁺	150 ns	5
p16(R24L)-CDK6 complex	71,899	85 × 98 × 103	3 Na ⁺	150 ns	5

Detailed concerns

[1] According to Figure 6A, in the models in the (starting?) conformation of the p16, R22, E9 and E12 are located far from each other. Based on this figure, similar conformation as with p15 would be possible just with slight side chain arrangement. Therefore, I am worried if the observed salt bridge / H-bond differences arise just based on the starting conformation of the simulations? This is not a minor point as this is used as a major explanation in the manuscript. This could be easily demonstrated by having good coverage of replicas starting for both conformations for p15 and p16. If these interactions are the definitive events, it should be seen in all simulations how p15 obtains the interactions regardless of the starting conformation and p16 loses (or do not obtain) these interactions with different starting conformations.

Response: Following the reviewer's suggestion, we checked the starting conformations of both CDK6-p15 and CDK6-p16 for MD simulations. As shown in Figure R1, key residues (R24 of p15, E18 and E21 of CDK6), that are found to be involved in CDK6-p15 interactions in MD simulations, are located far from each other in the initial structure, similar to the CDK6-p16 model. Therefore, for both initial structures of both CDK6-P16

and CDK6-P15 systems, the sidechains of key residues are far apart from each other. Interestingly for the CDK6-P15 system, these key interactions (salt-bridges and H-bonds) were found to be formed during the MD simulations. Following the reviewer's suggestions, we added up to 5 MD replicas for each model. The results are consistent with our previous finding that several salt-bridges can be formed for the CDK6-p15 model in MD simulations, but much less likely for the CDK6-p16 model, as shown in our revised Fig. 6A.

Figure R1, Analysis of key residue positions in the initial minimized structures of CDK6-p16 (colored cyan and blue) and CDK6-p15 (colored pink and magenta) before MD simulations. (A) superimposition of CDK6-p15 onto CDK6-p16, showing that the initial backbone conformations are similar to each other. (B) Check the key residue positions of CDK6-p16. (C) Key residue positions of CDK6-p15, showing that these key residues are not close to each other before MD simulations, similar to CDK6-p16 complex.

[2] The authors state that improved binding of p16(R24L) arises from the repulsion of R24 from this position. This is unclear if it would arise from repulsion of R24 or actually improved affinity arising from L24 (burying lipophilic residue from the solvent in the protein-protein interface usually increases binding affinity). Are there lipophilic contacts available for L24?

The observed interactions in the close proximity of the position 24, (for both R24 or L24), needs to be qualified and quantified. Also, what is the role for the starting conformation for the observed results as such short simulation was used?

Response: Following the reviewer's suggestion, we did further studies regarding this issue. In our original submission, we suggested that improved binding of p16^{R24L} arises from the salt-bridges formation of R22 (p16^{R24L}) with E18/E21 (CDK6), which could not be formed for p16 (maybe due to the repulsion force of the nearby R24 of p16 with R168 of CDK6). To provide further evidence to support this suggestion, MD simulations of CDK6-p16^{R24L} was performed, and the results showed that R22 of p16^{R24L} could partially recover the salt-bridges with E18 and E21 of CDK6. In our revised work, we added up to 5 MD replicas for

CDK6-p16^{R24L} model. The starting conformation of this complex is same as CDK6-p16 starting conformation. As shown in our revised Supplemental Figure 6B, in MD simulations, R22 in p16^{R24L} is likely to form salt bridges with E18 and E21 of CDK6, similar to R24 in p15, but much less likely for R22 in the wild-type p16.

In addition, following the reviewer's suggestion, we studied the interactions in the close proximity of the position 24 for both L24 and R24. All CDK6 residues involving in close interactions are analyzed by the distance distribution. The minimum distance of heavy atoms in MD simulations is used for distance analysis. The distance cutoff of 5 Å is used to select CDK6 residues. The *nativecontacts* command in *cpptraj* module of AmberTools^{1,2} is used for these analysis. For CDK6-p16^{R24L} model, as shown in Figure R2, R168 of CDK6 involves in close contact (minimum distance < 4 Å) with L24 of p16^{R24L} for 4 of 5 replicas. 2 of 5 replicas showed I169 of CDK6 interacts with L24 of p16^{R24L}. One replica showed L166 of CDK6 interacts with L24. The overall proximity environment of L24 is not hydrophobic. For CDK6-p16 model, as shown in Figure R3, 4 of 5 replicas showed stable close interactions (minimum distance < 3 Å) between D104 of CDK6 and R24 of p16. This analysis indicates that the improved binding affinity is not likely coming from Leucine.

Figure R2. The distance distributions of CDK6 residues that interacting with L24 of p16^{R24L}. The minimum distance of all heavy atoms between CDK6 residue and L24 of p16^{R24L} is used. All residues involving in minimum distance within 5 Å from L24 in MD simulations are shown here. For each MD replica, the last 100-ns trajectory is used for analysis.

Figure R3. The distance distributions of CDK6 residues that interacting with R24 of p16. The minimum distance of all heavy atoms between CDK6 residue and R24 of p16 is used. All residues involving in minimum distance within 5 Å from R24 in MD simulations are shown here. For each MD replica, the last 100-ns trajectory is used for analysis.

[3] Amino acids Glu9 and Glu12 of CDK6 are displayed to interact with p15. However, based on the CDK6 sequence (1-MEKDGLCRADQQ-12...) the residues in these positions are Ala9 and Gln12. Please assign the correct residue numbers.

Response: we thank the reviewer for catching this mistake. We have revised this in our manuscript. It should be amino acids Glu18 and Glu21 on the CDK6 sequence, as shown in Figure R1-C.

[4] The authors demonstrate the different deletions of N-terminal part reduces/abolishes binding to CDK4. However, all molecular modeling explanations are based on CDK6. How does this fit to your model? Although they are highly similar, please, demonstrate all similarities and dissimilarities of CDK4 and CDK6 on the putative p15/p16 interaction interfaces.

Response: We agree that all molecular modeling explanation are based on CDK6 (1. Based on the available crystallized CDK6-p16 as template to model CDK6-p15 complex; 2. Observed key interactions formation in MD simulations of CDK6-p15), which is related to the discussion about "Key Salt-bridges and Hydrogen-bonds between p15 and CDK6 Underlie Strong Binding" in the manuscript. Based on their highly similar sequences, we hypothesized that CDK4-p15 may have similar key

interactions. To experimentally examine these computational hypotheses, we carried out site-directed mutagenesis on p15 and followed by binding detection experiments (see Figure 6C and 6D) with CDK6, as well as CDK4. These results further substantiated these hypotheses.

Here, following the reviewer's suggestion, we demonstrated the sequence similarities and dissimilarities of CDK6 and CDK4 on the p16/p15 binding interface. As shown in Figure R4, based on sequence alignment of CDK4 and CDK6, residues on the putative p16/p15 binding interface are highlighted with orange background. The overall similarity for this interface between CDK4 and CDK6 is around 81%, and we identified that E21 of CDK6 is replaced with Valine in CDK4 (V14), and the nearby K26 of CDK6 is replaced with Threonine in CDK4 (T19). Considering that there is neither CDK4/P15 nor CDK4/P16 crystal structure available, to fully characterize CDK4/P15 and CDK4/P16 interactions would be beyond the scope of the current work. We plan to address it in our future work.

Figure R4. Sequence alignment of human CDK4 and CDK6. All the residues on the putative p16/p15 binding interface (within 5 Å of p16/p15) are highlighted with orange background.

[5] The authors are referring to the wrong supplementary fig. in: "We identified several key interactive points between the N-terminus of p15 and CDK6, particularly in the salt-bridges and H-bonds (Fig. 6A; Supplementary Fig. 5A-D)"

Response: We have revised this in our manuscript.

[6] In supplementary figures, are the shown H-bond probability towards side chain, backbone or both? This is crucial as the authors are stating is as a salt bridge. Please differentiate clearly

backbone H-bonds and salt bridges in the text and in the figures.

Response: Instead of using previous H-bond probability analysis, we carried out distance distribution analysis for key residue interactions in our revised manuscript. The distance distribution for each pair of residues is calculated by the minimum distance among all atoms (including hydrogen atoms) of the pair. The *nativecontacts* command in *cptraj* module of AmberTools^{1,2} is used for these analysis. For each pair of residues, we compared the distance distributions among CDK6-p15, CDK6-p16 and CDK6-p16^{R24L} models. Figure R5 illustrates the average distance distribution overall all 5 MD replicas (combined 500-ns trajectories) for different residue pairs. And a detailed analysis regarding to each replica is shown in Figure R6.

As shown in Figure R5, for CDK6-p15 model, Arg24 (p15) – Glu18 (CDK6), Arg45 (p15) – Glu18 (CDK6), Arg24 (p15) – Glu21 (CDK6), Glu16 (p15) – Arg45 (p15) and Ser20 (CDK6) – Glu18 (CDK6) form stable close interactions in MD simulations. Three salt-bridges, Arg24 (p15) – Glu18 (CDK6), Arg45 (p15) – Glu18 (CDK6) and Arg24 (p15) – Glu21 (CDK6), and a hydrogen bond, Ser20 (CDK6) – Glu18 (CDK6), are formed in CDK6-p15.

CDK6-p16 model do not obtain these stable salt-bridge formation (Figure R5-A,B,C), while two salt-bridges, Arg22 (p16) – Glu18 (CDK6) and Arg22 (p16) – Glu21 (CDK6), are formed in CDK6-p16^{R24L} model. In Figure R5-D, Glu16 (p15) – Arg45 (p15) forms stable salt-bridge in p15, this similar interaction, Asp14 (p16) – Ser43 (p16), is also observed p16 and p16^{R24L}, but less stable. In Figure R5-E, we can see that all three models obtain close interactions between Ser20 (p15)/Thr18 (p16) and Glu18 (CDK6). We updated these results in our revised manuscript.

Figure R5. The distance distributions of 5 key interactions for CDK6-p15, compared with CDK6-p16 and CDK6-p16^{R24L}. For each pair of residues, the average minimum distance distribution overall all 5 MD replicas (combined 500-ns trajectory) is calculated.

Figure R6. The distance distributions of 5 key interactions for CDK6-p15, compared with CDK6-p16 and CDK6-p16^{R24L} in different MD replicas. Each plot shows 5 distance distributions from 5 replicas for one pair of residues.

[7] In the methods, a clustering approach was used for “binding interface analysis”. First, I am puzzled why the authors applied such a method for these short trajectories? Furthermore, it is unclear in the manuscript where this method was applied, please clarify. I suspect that this type of approach will only further bias already biased (under sampled) data.

Response: In our original submission, DBSCAN clustering approach^{3,4} is applied to cluster these trajectories to obtain representative structure, which we have shown for CDK6-p15 key interactions in Figure 6A. In our revised manuscript, based on the updated 5 MD replicas (a combined 500-ns trajectories, which have 5000 snapshots with 100 ps interval), we accordingly updated the clustering results and add descriptions in the manuscript. It should be noted that clustering analysis is only one of several aspects of our structural analysis. We also analyzed the minimum distance distribution of key interactions in all replicas, as shown in Figure R5 and Figure R6.

[8] The overall stability and/or instability of the interacting protein, needs to be quantitatively described. For this, e.g. RMSF values would be useful.

Response: Following the suggestion, we have incorporated B-factor analysis in Supplemental Figures. As shown in Figure R7, for p15, p16 and p16^{R24L} proteins that interact with CDK6, they are relatively stable in MD simulations, except for the intrinsically disordered N-terminal (1-11) and the C-terminal. For CDK6, the B-factor analysis shows consistently flexible region and stable region in these three systems. The most flexible region of CDK6 in sequence is 170-176, which is a loop region called T-loop, it towards water environment and can be phosphorylation by other partners. Other flexible region of CDK6 (47-60, 86-91, 240-258) are far from this protein-protein interacting interface.

Figure R7. B-factor analysis of CDK6-p15, CDK6-p16 and CDK6-p16^{R24L} complexes in MD simulations.

[9] The mouse and human p15 are not totally identical (shorter N-terminal in mouse and some amino acid differences). How does this reflect to your modelling explanation of the observed interactions?

Response: Our modeling studies focus on human p15 and aim to explain those results related to human p15 (such as the corresponding binding detection experiments of Figure 6C and 6D in original manuscript). In addition, sequence alignment between human p15 and mouse p15 are shown in Figure R8, the differences mainly come from N-terminal residues 1-10, which are intrinsically disordered segment and not involved in key interactions formation for CDK6-p15(human) model. For residues involved in key interactions with CDK6, E16 of human p15 is

replaced with Ala in mouse p15, S20 of human p15 is replaced with Thr in mouse p15. Other key residues (R24, R45) are the same. The overall structure should be similar for human p15 and mouse p15, except for the N-terminal (1-10) difference.

Figure R8. Sequence alignment of human and mouse p15.

[10] In supplementary figure 10 RMSD values of systems are presented. Which RMSD is this, protein backbone or does it include also sidechains or what? Please define.

Response: The RMSD of supplementary figure 10 is calculated by protein C_alpha atoms. In our revised manuscript, we revised it accordingly in the figure caption.

[11] Describe how replicas differ from each other, according to the system setup the initial configuration was identical. I was unable to locate this information from the manuscript.

Response: All replicas are indeed identical in the starting structural configuration, but with different initial velocities in MD simulations. We have clarified it in our manuscript.

[12] I wonder why such a small timesteps as 5 ps were used for the analysis? "The atomic coordinates of the complexes were saved every 5 ps to obtain the trajectories for post structural analysis"

Response: In our revised work, number of replicas added up to 5 for each system. For each system, a combined 500-ns trajectories (5 replicas of last 100-ns trajectories) were analyzed with the interval of 100 ps, that is 5000 snapshots for post structural analysis. We have accordingly revised this in the manuscript.

[13] The authors have not uploaded the raw simulation data readily available to one of the public repositories. This is highly recommended practice in the field and provides further confidence for the simulation data3.

3 Abraham M et al.: Sharing Data from Molecular Simulations. J Chem Inf Mod 2019, 59, 10, 4093–4099

Response: Thanks for the reviewer's kind suggestion. Total raw simulation data is around 380 GB, which would be too large for uploading to public repositories. For each MD replica, the whole production simulation data without waters/ions and with an interval of 100 ps has been prepared and uploaded to: https://www.nyu.edu/projects/yzhang/MD_data .

REVIEWER COMMENTS

Reviewer #4 (Remarks to the Author):

The authors have conducted nice revision work to address my concerns and only the following minor points remain:

- 1) Define the “top” in the sentence p.22: “The top p15 model built based on p16 was selected and superimposed on the p16-CDK6 complex to build p15-CDK6 complex (Supplementary Fig. 5D).”
- 2) Please include information to the methods how B-factor analysis was conducted (which method etc.?). The B-factor SI figure is also missing y-axis label.
- 3) Differences among RAS mutants has become more and more evident during recent years and these mutant specificities are currently under active research in the field. Therefore, I suggest that the authors would change the general term: “oncogenic HRasWT/*” that is used in the manuscript to more specific ” HRasWT/Q61L ”, which describes the mutation context more clearly.

Typo in Figure 6 text. “Molecular dynamic simulations” → Molecular dynamics simulations

Reviewer #4 (Remarks to the Author):

The authors have conducted nice revision work to address my concerns and only the following minor points remain:

1) Define the “top” in the sentence p.22: “The top p15 model built based on p16 was selected and superimposed on the p16-CDK6 complex to build p15-CDK6 complex (Supplementary Fig. 5D).”

Response: The “top” means the most confident homology model among five models predicted by I-TASSER. In I-TASSER, it reports up to five models and the confidence of each model is quantitatively measured by C-score. A C-score of higher value signifies a model with a higher confidence. In our modelling, we chose the p15 model with the highest C-score. Following the reviewer’s suggestion, we revised the sentence to the following:

“Among five p15 models predicted by I-TASSER based on p16, the one with the highest C-score was selected and superimposed on the p16-CDK6 complex to build p15-CDK6 complex (Supplementary Fig. 5D).”

2) Please include information to the methods how B-factor analysis was conducted (which method etc.?). The B-factor SI figure is also missing y-axis label.

Response: Thanks for the reviewer’s careful reading. We revised it accordingly in the manuscript for the methods section as well as for the B-factor SI figure.

“ **Post structural analysis of MD trajectory.** The root-mean-square deviation (RMSD) profiles of protein C alpha atoms (Supplementary Fig. 10A) and B-factor analysis (Supplementary Fig. 10B) for p15-CDK6, p16-CDK6 and p16^{R24L}-CDK6 complexes in different MD replicas were performed. For each MD replica, the residue B-factor analysis was conducted on the last 50-ns trajectories using *atomicfluct* command in *cpptraj* module of *AmberTools*⁶¹. The residue B-factor is the mass-weighted average of positional fluctuations squared of all atoms in each residue multiplied by a constant $8\pi^2/3$.”

3) Differences among RAS mutants has become more and more evident during recent years

and these mutant specificities are currently under active research in the field. Therefore, I suggest that the authors would change the general term: "oncogenic HRasWT/*" that is used in the manuscript to more specific " HRasWT/Q61L ", which describes the mutation context more clearly.

Response: We appreciate the Reviewer's suggestion and have specified the HRas we used as Q61L in both the Methods and Results sections when the term was first mentioned.

Typo in Figure 6 text. "Molecular dynamic simulations" -> Molecular dynamics simulations

Response: we thank the reviewer for catching this typo. We have revised this in our manuscript.